# Prospective Climates, and Water Availabilities under Different Projections of Environmental Changes in Prince Edward Island, Canada

**Ahmad Zeeshan Bhatti** [1,*], **Aitazaz Ahsan Farooque** [1,2,*], **Nicholas Krouglicof** [1], **Wayne Peters** [1], **Qing Li** [3] **and Bishnu Acharya** [4]

1   Faculty of Sustainable Design Engineering/Science, University of Prince Edward Island, Charlottetown, PE C1A 4P3, Canada; nkrouglicof@upei.ca (N.K.); wpeters@upei.ca (W.P.)
2   School of Climate Change and Adaptation, University of Prince Edward Island, Charlottetown, PE C1A 4P3, Canada
3   Department of Environment, Energy, and Climate Action, Government of Prince Edward Island, Charlottetown, PE C1A 7N8, Canada; qli@gov.pe.ca
4   Department of Chemical and Biological Engineering, University of Saskatchewan, Saskatoon, SK S7N 5A9, Canada; bsp874@mail.usask.ca
*   Correspondence: azbhatti@upei.ca (A.Z.B.); afarooque@upei.ca (A.A.F.); Tel.: +1-(902)-566-6084 (A.A.F.)

**Abstract:** Climate change impacts on temperatures, precipitations, streamflows, and recharges were studied across eastern, central, and western Prince Edward Island (PEI) between climate normals in 1991–2020, 2021–2050, and 2051–2080 using observed and projected data, and SWAT modeling. Average annual temperature can significantly rise from the existing 5.90–6.86 °C to 8.26–11.09 °C in different parts during the next 30–60 years under different RCP scenarios. Average annual precipitations would not significantly change except in western PEI where a 17% likely increase would offset further warming impact; therefore, current streamflows (~650 mm/year) and recharges (~320 mm/year) would not be much affected there. However, warming and increased pumping together in its Wilmot River watershed could reduce streamflows up to 9%, and 13% during 2021–2050, and 2051–2080, respectively. In the eastern forest-dominated Bear River watershed, no significant reductions in current streamflows (~692 mm/year) or recharges (~597 mm/year) are expected. Nevertheless, near constant precipitation and warming could cumulatively reduce streamflows/recharges up to 8% there, as pumping will be negligible. In the central zone, precipitation could insignificantly increase up to 5%, but current streamflows (~737 mm/year) and recharges (~446 mm/year) would not be significantly affected, except for RCP8.5 under which streamflows could reduce by ~16% during 2051–2080. Overall, more attenuated streamflows and recharges are likely with higher quantities in late winter and early spring, and somewhat lesser ones in summer, which could reduce water supplies during the growing season. Besides, precipitation uncertainty of ~300 mm/year between dry and wet years continues to be a major water management challenge. Adapting policies and regulations to the changing environment would ensure sustainable water management in PEI.

**Keywords:** climate change; Canada; temperature; precipitation; streamflows; groundwater recharge; water availability

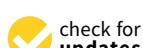



## 1. Introduction

Direct and indirect human activities continue to alter the composition of the global atmosphere by increasing the concentration of greenhouse gases (GHG). The Intergovernmental Panel on Climate Change (IPCC) projects future climates based on human GHG emissions from all sources. Different scenarios are presented, expressed in terms of radiative forces (watts/m$^2$) expected by the year 2100. The scenarios are called Representative

Concentration Pathways (RCPs), e.g., RCP2.6, RCP4.5, and RCP8.5. RCP2.6 targets to keep global warming below 2 °C above pre-industrial temperatures by 2100. That will, however, require substantial net negative emissions. RCP4.5 is an intermediate scenario, but also a net negative emissions one, which projects to keep global temperature rise between 2 and 3 °C, whereas RCP8.5 assumes emissions will rise throughout the 21st century with a projected increase in global temperatures up to 5 °C [1,2].

Water resources are particularly affected by climate change [3,4]. Streamflows and groundwater recharges, as well as their spatial and temporal distribution changes caused by changes in temperature, evaporation, and precipitation [5]. Both hydrological components (streamflows and groundwater recharges) have been impacted worldwide because of changes in climatic components, i.e., rises in temperatures and changed precipitation patterns [6,7] and they will continue to be impacted. A global hydrological assessment reveals that by 2050, hot and dry regions near the mid-latitudes would become drier and that average annual streamflows are likely to drop by 10–30%. In contrast, wet regions at high latitudes may become wetter and streamflows may increase by 10–40% there [8], and the trend would extrapolate by 2100. The case of groundwater recharge would be similar. Nevertheless, different case studies find high spatiotemporal variations in the hydrological impacts of climate change, such as the streamflows of the rivers originating from the Hindukush-Himalaya (HKH) region, i.e., the Indus, Ganges, Brahmaputra, Salween and Mekong rivers, which have been projected to increase up till 2050 because of increased precipitation and accelerated snowmelt [9]. In the Heihe River Basin of China, increased precipitation has increased streamflows in the last 50 years, whereas land use and potential evapotranspiration (PET) changes showed minimalistic impacts [10]. In contrast, streamflows in the Soan River Basin of Pakistan, a tributary of the River Indus, has been significantly reduced since 1997 because of changed precipitation and PET, and somewhat because of land-use changes [11]. Besides quantitative changes, temporal redistribution of streamflows and groundwater recharges under climate change would also affect water availabilities in different seasons. For example, earlier spring snowmelt would cause North American rivers to run 30–40 days earlier and, therefore, they would need adapted water management [12]. Furthermore, population growth, increased per capita water demands, and environmental pollution will increase stress on water resources [3]. Therefore, water managers should consider the prospective environments for sustainable planning and management of water resources [13].

Canadian water resources are also impacted by climate change in various aspects. Warming here was 0.6 °C higher than the global average of 1 °C during the 20th century. During the same timescale, average precipitation increased from under 500 mm/year to about 600 mm/year. However, it spatially varied from −10% in the drier Prairies to +35% in the northern and southern parts. Streamflows and groundwater recharge mostly exhibited stability under the climatic changes with some temporal redistribution. In general, early streamflow peaks and increased winter flows were observed. Accordingly, more groundwater recharge was observed in late winter and spring because of reduced frost cover and increased infiltration, and somewhat less in summer because of more intense rainfall events and more evaporation [14,15].

Prince Edward Island (PEI) is an Atlantic Canada province, located between latitudes 45°57′ and 47°04′ N and longitudes 61°55′ and 64°25′ W in the Gulf of Saint Lawrence (Figure 1). It is the smallest (5750 km$^2$ area) and most populous (27 persons/km$^2$) province. Groundwater provides a high contribution of up to 73% to streamflows and, therefore, the province is entirely dependent on groundwater for all types of requirements. The climate is cool and humid with long, mild winters and moderately warm summers. Annual precipitation is about 1100 mm, of which 80 percent is rainfall and the rest is snowfall. Precipitation is evenly distributed throughout the year [16–19]. Climate change impacts are evident in terms of warming, changed precipitation patterns, and the hydrology thereof. A latest analysis reveals statistically significant warming on the island, wherein average annual temperatures increased by 1.14 °C in the east and 0.75 °C in the west during 1991–2020

compared to those during 1961–1990. The warming trend was distributed throughout the year, and most months showed statistically significant warming. Proportionately, the average minimum temperature rose more than the maximum, thus lowering the cold intensity. Comparative precipitation analysis between the two periods reveals that precipitation increased by 6% in the east and decreased by 5% and 8% in central and western PEI, respectively. Moreover, rainfall intensities were found to increase, as well, by 2 mm/h on average [19].

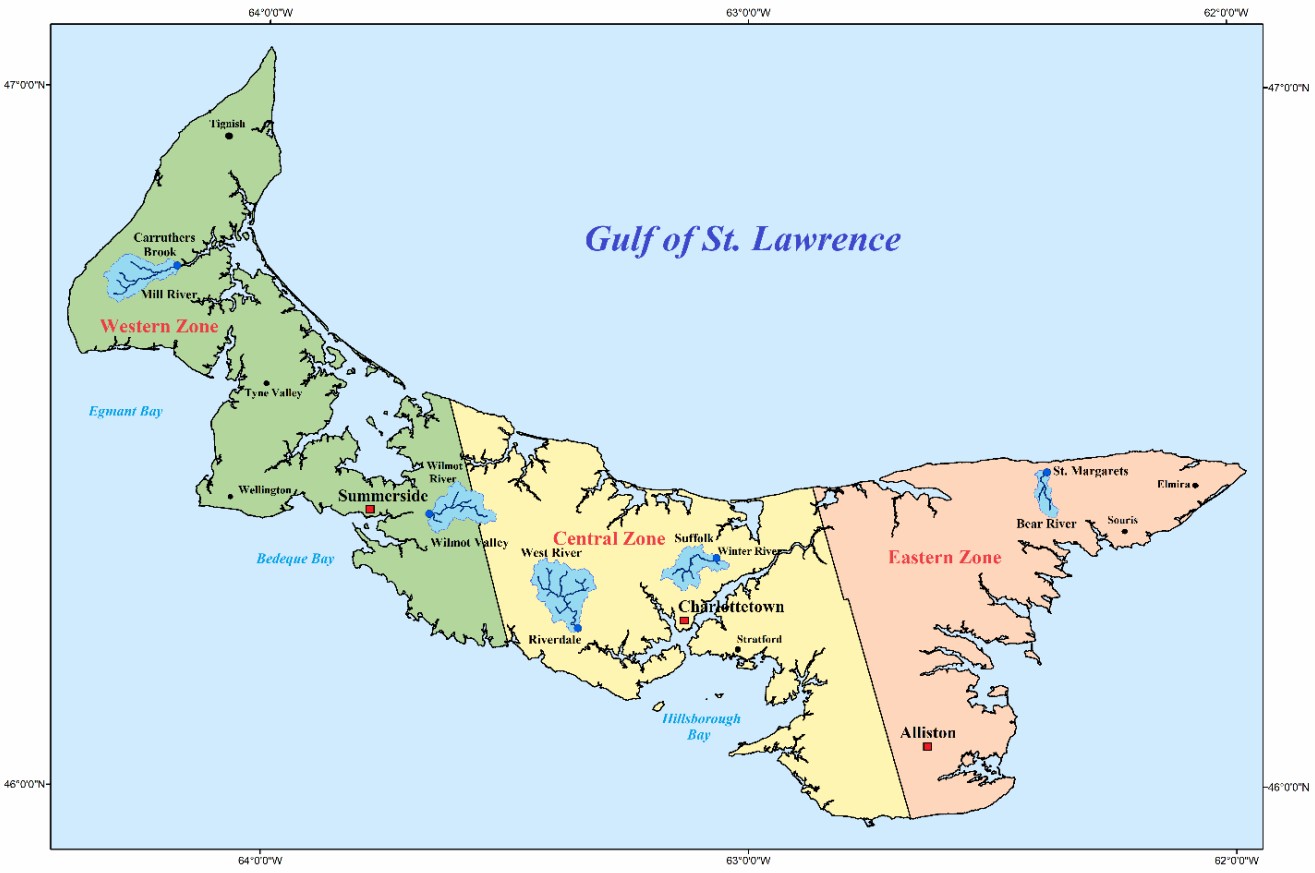

**Figure 1.** PEI climatic zones, weather stations ■, and modeled watersheds ▫.

Streamflow and groundwater recharge assessments were done during the periods 1995–2004 and 2005–2014 in different watersheds across PEI. It was found that average annual streamflows in the watersheds of the Mill, Wilmot, West, Winter, and Bear rivers, at their respective monitoring points (Carruthers Brook, Wilmot Valley, Riverdale, Suffolk, and St. Margaret's), during 1995–2004 were 596, 598, 747, 505, 737 mm/year respectively, which subsequently increased to 680, 708, 891, 601, and 799 mm/year, respectively, during 2005–2014. This indicates an increase of 9 to 19% in the average annual streamflows of those watersheds over the 2 decades. The average annual recharges in those watersheds during 1995–2014 were 314, 345, 419, 504, and 600 mm/year in the Mill, Wilmot, West, Winter, and Bear River watersheds, respectively [20,21]. The average annual recharge in the whole of PEI was estimated to be about 407 mm/year, whereas Li [22] estimated it as 424 mm/year during 1961–1990, which was predicted to increase by 7%, 9%, and 14% by the 2050s (2041–2070) under the RCP2.6, RCP4.5, and RCP8.5 scenarios, respectively [22].

The study is an in-depth investigation of changes in temperature, precipitation, streamflows and groundwater recharges across PEI under different projections of environmental changes. Spatially, the analysis is divided into the eastern, central, and western zones (Figure 1) and temporally into climate normals: 1991–2020 (2000s), 2021–2050 (2030s), and

2051–2080 (2060s). The 30-year analytical periods were adopted for comparative analysis as per the guidelines of the World Meteorological Organization (WMO), which specifically defines a climate normal as, "Period averages computed for a uniform and relatively long period comprising at least three consecutive ten-year period [23] and a standard practice in North America" [24]. The following research questions have been answered: (i) how the average monthly and annual temperatures that prevailed during 1991–2020 would change in different regions of PEI during the 2030s (2021–2050) and 2060s (2051–2080) under RCP scenarios 2.6, 4.5, and 8.5; (ii) what the expected monthly and annual precipitations in normal, dry, and wet years in different regions of PEI during the 2030s (2021–2050) and 2060s (2051–2080) under the RCP scenarios would be; (iii) how the average monthly and annual precipitations that prevailed during 1991–2020 would change in different regions of PEI during the 2030s (2021–2050) and 2060s (2051–2080) under the RCP scenarios; (iv) what the streamflows and groundwater recharges in the eastern, central, and western zones of PEI during the 2030s (2021–2050) and 2060s (2051–2080) under the RCP scenarios would be; and (v) whether the demarcated climatic and hydrological changes are statistically significant.

## 2. Materials and Methods

The prospective climate change analysis has been spatially segregated into the eastern, central, and western zones, which administratively correspond respectively to the King, Queen, and Prince counties of PEI (Figure 1). Climatic stations at Alliston, Charlottetown, and Summerside represent the eastern, central, and western zones, respectively. Similarly, the Bear River watershed represents the eastern zone, the West and Winter River watersheds represent the central, and the Mill and Wilmot River watersheds represent the western zone. Temporally, 30-year analytical periods (climate normal) were adopted as per WMO guidelines [23]. Historical data of the latest climate normal in the 2000s (1991–2020) were compared with the projections for 2021–2050 (2030s), and 2051–2080 (2060s) under RCP scenarios 2.6, 4.5, and 8.5. Several worldwide studies used historical and modeled projected climatic and hydrological data to demarcate the impacts of climate change for better water management [10,11,19,25–28].

Pacific Climate Impacts Consortium (PCIC) data has been used for future projections, which provide statistically downscaled daily precipitation, minimum, and maximum temperatures at 10 km × 10 km resolution under different RCP scenarios. The PCIC's data credibility in PEI was verified by Bhatti et al. [19], who compared observed mean monthly precipitation, mean monthly maximum, and mean monthly minimum temperatures during 1943–2020, with the PCIC's data -downscaled by Bias Correction/Constructed Analogues with Quantile (BCCAQ) method for RCP4.5 at Alliston, Charlottetown, and Summerside and found it appropriate to be used for climate change analysis [19]. Li [22] used futuristic PCIC data (2041–2070) for modeling the different watersheds of PEI using HEC-HMS 4.1 under the RCP2.6, RCP4.5, and RCP8.5 scenarios, and estimated streamflows and groundwater recharges accordingly [22]. Daraio et al. [29] also used the BCCAQ technique to downscale PCIC data for flood mapping and infrastructure design in the neighboring Newfoundland and Labrador province [29]. PCIC data were therefore found appropriate and used for prospective climatic and hydrological projections of the study. The methodologies adopted for precipitation, temperature, streamflows, and groundwater recharge projections are given in the subsequent sections.

### 2.1. Precipitation

Precipitation patterns have been analyzed by developing yearly and monthly Probability of Exceedance (PoE) curves, for different climate normals (2000s, 2030s, and 2050s), at the three climate stations (Alliston, Charlottetown, and Summerside) [30]. PoE is a popular method used to determine precipitation expectancy in an area for normal (PoE = 50%), wet (PoE = 20%), and dry (PoE = 80%) years in a 30-year period (climate normal). Precipitation data for the analytic period 1991–2020 were the observed data, ascertained from the respective climate station/observatory, whereas those for 2021–2050 and 2051–2080 were PCIC

projections under either the RCP2.6, 4.5, or 8.5 scenario. The observed and PCIC's daily precipitations were added to get monthly and yearly precipitations for all the analytical periods, i.e., 1991–2020 (2000s), 2021–2050 (2030s), and 2051–2080 (2060s) [24]. The yearly or monthly precipitations for each of the 30-year periods of climate normal were arranged in descending order and assigned ranks (r). The year with the highest precipitation has r = 1 and that with least has r = 30. The standard formula followed by the World Meteorological Organization (WMO) was used to compute that PoEs = $(r - 0.44)/(n + 0.12) \times 100$, where $n = 30$ (number of years in a climate normal). Precipitations were plotted against respective PoEs, and a linear curve was imposed by removing four to five outliers at the edges of the curves to have the best fit lines having a coefficient of determination above 90% [30]. Yearly and monthly expected precipitations for normal, wet, and dry years were extracted from the curves and compared for the different climate normals, RCP scenarios, at the three locations. Accordingly, Doria and Madramootoo [26] used PoEs to estimate future irrigation requirements in southern Quebec, Canada under dry, normal, and wet years [26], and by Bhatti et al. [19] in PEI. The results were integrated, presented, and discussed to better grasp the prospective changes in precipitation expectancy.

The average annual and monthly precipitations during the three climate normals were computed from the daily data by simply getting the arithmetic means of the yearly or monthly totals for each of the 30-year periods [23,24]. Changes in precipitations during the 2030s and 2060s under different RCP scenarios were discussed in comparison to those that prevailed during 1991–2020. The statistical significances of the yearly and monthly means were determined by applying a one-way analysis of variance (ANOVA) between the climate normals 1991–2020, 2021–2050, and 2051–2080, wherein the latter two normals have a further three scenarios, i.e., RCP2.6, 4.5, and 8.5. For this purpose, a Minitab 20 (Pennsylvania State University, State College, PA, USA, Minitab, Inc.) was used for ANOVA and multiple means comparisons tests. Prior assumptions of the test, that is, independent observations, normal distribution, and within-groups homoscedasticity, were verified, and the means were compared with Tukey's pairwise comparisons test at 95% confidence $\alpha = 0.05$ [19,31].

### 2.2. Temperature

Daily minimum and maximum temperatures measured at the three climate stations were available during the period 1991–2020. Similarly, future projections by the PCIC provided daily minimum and maximum temperatures for the periods 2021–2050 and 2051–2080 under different RCP scenarios. Mean daily temperatures were ascertained by simply averaging the daily maximum and minimum temperatures and mean monthly temperature as an average of all the mean daily temperatures as a standard North American practice [32]. Therefore, average monthly and annual temperatures during the three climate normals were the average of mean monthly and mean yearly temperatures for each of the 30-year periods, as per Karl et al. [32] and Bhatti et al. [19]. Average monthly temperatures under different RCP projections were plotted against the observed temperatures during 1991–2020, separately for the three stations. Moreover, the statistical significances of the monthly and yearly means were determined between the three climate normals and projections by the One-Way ANOVA, same as for the precipitation analysis [19,31]. Garbary [33] also used the same techniques to assess annual and monthly warming in PEI over the periods 1961–1990 and 1991–2016 [33].

### 2.3. Streamflows

Prospective streamflow and groundwater recharge assessments were done for different watersheds scattered across PEI, i.e., the Mill, Wilmot, West, Winter, and Bear River watersheds, as shown in Figure 1. Distributed and semi-distributed hydrological models have been used around the world to demarcate the impacts of climate change on streamflows, ET, etc. [10,11]. The Soil and Water Assessment Tool (SWAT) is a semi-distributed, continuous-time model that integrates topographical, climatic, soil, and landcover data

to simulate watershed hydrology, and has been used worldwide to demarcate impacts of climate change [34–37]. Sood et al. [25] used it to simulate the climate change impacts on hydrology in West Africa for 2021–2050 and 2071–2100, using 1983–2012 as the reference period [25]. Accordingly, a validated SWAT model for the same watersheds was available [20]. It was previously calibrated against observed mean monthly flows (m³/s) from 1995–2004, and validated for 2005–2014, monitored at the points as detailed in Table 1. The model successfully simulated mean monthly flows, excluding pumping effects, as per the accuracy criteria i.e., $R^2$ = 0.5–1.0, NSE = 0.5–1.0, RSR <= 0.7, and PBIAS ± 25%, wherein NSE carried the highest weight being snowmelt-oriented watersheds. SWAT-CUP's SUFI-2 procedure was used for parameter sensitivity analysis, and multisite calibration. Parameters were adjusted during calibration because of strong surfacewater-groundwater interactions, where the baseflow contribution to streamflows is as high as 73% [20].

**Table 1.** The modeled watersheds and their streamflow monitoring points in PEI.

| Watershed | Monitoring Point | Station No. | Latitude | Longitude | Drainage Area (km²) |
|---|---|---|---|---|---|
| Mill River | Carruthers Brook | 01CA003 | 46.744 | −64.187 | 48.7 |
| Wilmot River | Wilmot Valley | 01CB004 | 46.393 | −63.659 | 45.9 |
| West River | Riverdale | 01CC005 | 46.231 | −63.351 | 70.1 |
| Winter River | Suffolk | 01CC002 | 46.332 | −63.065 | 36.2 |
| Bear River | St. Margaret's | 01CD005 | 46.453 | −62.382 | 15.4 |

The validated SWAT model was simulated for 2021–2080 under the RCP2.6, 4.5, and 8.5 scenarios. Futuristic simulations under each RCP scenario required updated precipitation (pcp) and temperature (tmp) input files. The pcp files contained daily precipitations (mm) from 1 January 2021 to 31 December 2080, and the tmp file contained daily maximum and minimum temperatures (°C) for the same durations at the six stations (Table 2). The data were acquired from PCIC, downscaled under the BCCAQ method for the 10 km × 10 km grid centered around the Weather Generator (WGEN) stations, i.e., 464–631, 464–634, 467–644, 467–641, 464–625, and 464–638, as detailed in Table 2. Prior to each of the three simulations, the respective pcp and tmp files of the six stations were replaced in the root folder of the validated model without disturbing any other parameter and simulated accordingly to get the results [20].

**Table 2.** WGEN stations used in modeling located in close proximity of the five watersheds.

| WGEN Station | Proximity | Zone | Latitude | Longitude | Elevation (m) |
|---|---|---|---|---|---|
| 464–631 | Winter River | Central | 46.40 | −63.10 | 40 |
| 464–634 | West River | Central | 46.40 | −63.40 | 96 |
| 467–644 | Mill River | Western | 46.70 | −64.40 | 44 |
| 467–641 | Mill River | Western | 46.70 | −64.10 | 19 |
| 464–625 | Bear River | Eastern | 46.40 | −62.50 | 30 |
| 464–638 | Wilmot River | Western | 46.40 | −63.80 | 01 |

The model was simulated on a daily time step; however, the results were generated in terms of mean monthly flows (MMF) in m³/s. This is how 30 MMFs were ascertained for each month of the year and for each duration: 2021–2050 and 2051–2080. Average monthly flows (m³/s) were computed by getting the arithmetic means of the respective 30 MMFs belonging to 2021–2050 and 2051–2080. Mathematically, $Q_{avg} = \frac{\sum_{i=1}^{30} MMF_i}{30}$, where $Q_{avg}$ is the average monthly flow either during 2021–2050 or 2051–2080, and $MMF_i$ are the respective mean monthly flows of the *i*th year, both expressed in m³/s [23]. Simulated average monthly flows for 2021–2050 and 2051–2080 were ascertained for all the RCP2.6, 4.5, and 8.5 scenarios. The results were, however, converted to mm to improve the un-

derstandability of the water balance across zones and watersheds and were statistically analyzed to find significant differences across climate normals and RCP scenarios [19,31].

It was found during validation that the model overpredicted flows equal to that of pumpings in respective watersheds, as SWAT cannot account for pumping. For instance, the average annual observed flow at Suffolk in the Winter River watershed during 2005–2014 was 555 mm/year, whereas the simulated flow during that period was 779 mm/year; the difference is close to pumping, i.e., 191 mm/year [20], and in the purview of the above, it is assumed that pumping would reduce the same amount of streamflows on an annual basis in terms of the water mass balance. Groundwater pumpings will increase in future under the rising sectoral demands; therefore, future scenarios were developed considering increased pumpings. Simulated average monthly flows were accordingly adjusted under the likely pumpings. The anticipated pumpings and adjustments made in the simulated streamflows thereof under different RCP scenarios are given in Table 3.

**Table 3.** Anticipated pumpings and adjustments made in simulated streamflows (mm/year).

| Watershed | Hydrological Parameter | 1995–2014 | 2021–2050 | | | 2051–2080 | | |
|---|---|---|---|---|---|---|---|---|
| | | Modeled | RCP2.6 | RCP4.5 | RCP8.5 | RCP2.6 | RCP4.5 | RCP8.5 |
| Mill River | Pumping | 1 | 1.30 | 1.40 | 1.50 | 1.60 | 1.70 | 1.80 |
| | Streamflow | 643 | −1.30 | −1.40 | −1.50 | −1.60 | −1.70 | −1.80 |
| Wilmot River | Pumping | 16 | 20.80 | 22.40 | 24.00 | 25.60 | 27.20 | 28.80 |
| | Streamflow | 652 | −20.80 | −22.40 | −24.00 | −25.60 | −27.20 | −28.80 |
| West River | Pumping | 1 | 1.30 | 1.40 | 1.50 | 1.60 | 1.70 | 1.80 |
| | Streamflow | 811 | −1.30 | −1.40 | −1.50 | −1.60 | −1.70 | −1.80 |
| Winter River | Pumping | 191 | 191 | 191 | 191 | 191 | 191 | 191 |
| | Streamflow | 555 | −191 | −191 | −191 | −191 | −191 | −191 |
| Bear River | Pumping | 1 | 1.30 | 1.40 | 1.50 | 1.60 | 1.70 | 1.80 |
| | Streamflow | 691 | −1.30 | −1.40 | −1.50 | −1.60 | −1.70 | −1.80 |

Col. 3 Source: Bhatti et al. [20].

It was assumed that pumpings would increase in all the watersheds, except the Winter River's, by an additional 30% during 2021–2050, and by 60% during 2051–2080 under the RCP2.6 scenario. Since hotter weather is likely to further increase water requirements and the pumping thereof for all sectors, an additional increment of 10% was assumed for the RCP4.5 scenario, and 20% for the RCP8.5 scenario. High water stress in the upper Winter River watershed has been sensed and managed by the government. Therefore, large groundwater extraction permits won't be issued in the Winter River watershed. The City of Charlottetown, the single largest user of pumped groundwater in this watershed, is decreasing pumping and getting more water from other watersheds. In the purview of the trend, it has been assumed that pumping would remain the same in this watershed [22]. As evident from the table, that pumping would affect streamflows in the Wilmot and in Winter River watersheds; therefore, pumping-adjusted results have been restricted to those two watersheds. The results were discussed along with the previous findings, reasons, and impacts of probable changes, and mitigation strategies.

### 2.4. Groundwater Recharges

Groundwater recharges were assessed in the five watersheds using the same SWAT simulations as for the streamflow assessments (Section 2.3). SWAT generates groundwater recharge at the subbasin scale, and there were 53 subbasins in the model as shown in Figure 2. The westernmost Mill River watershed has seven subbasins (1–7), the easternmost Bear River has three (8, 9, 18), and similarly for the other watersheds as depicted in the Figure. The "Output" file of the simulated model contained all hydrological parameters' values including groundwater recharges (mm) on a monthly timescale for all the 53 subbasins. The results were spatially and temporally aggregated for better understanding.

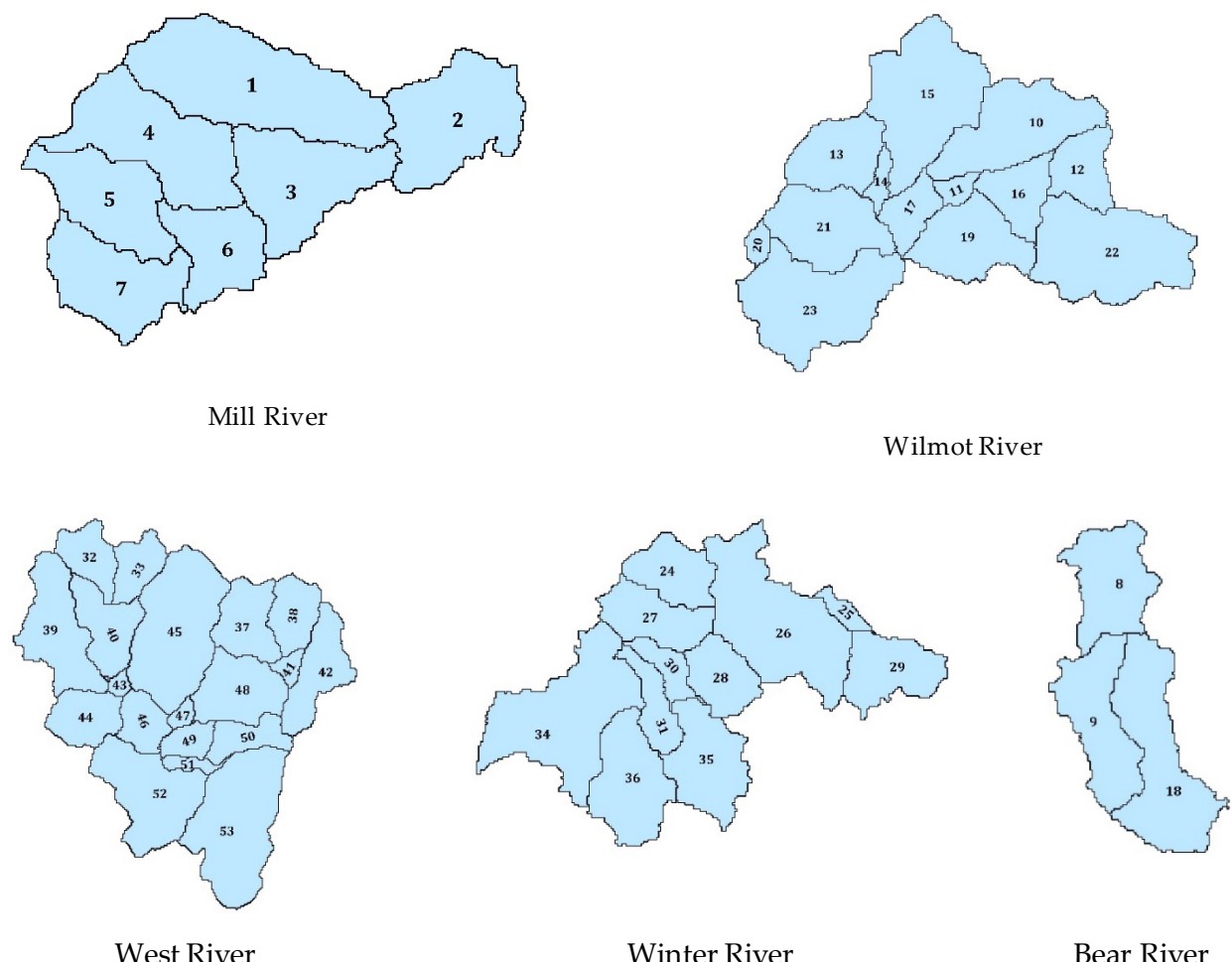

**Figure 2.** Subbasins of different watersheds modeled by SWAT.

Temporally, for each subbasin, 30 monthly recharges for each month of a year were separately averaged for 2021–2050, and for 2051–2080. Mathematically, $R_{avg} = \frac{\sum_{i=1}^{30} MR_i}{30}$, where $R_{avg}$ is the average recharge of a month in a subbasin either during 2021–2050 or 2051–2080, and $MR_i$ is the recharge of a month in the $i$th year of either 2021–2050 or 2051–2080, both expressed in mm [23]. The ascertained average recharges of the months were spatially aggregated for each watershed, by taking the weighted average of different subbasins in a watershed, e.g., $R_{Mill\ River} = \frac{\sum_{i=1}^{7} R_i \times A_i}{\sum_{i=1}^{7} A_i}$, where $R_{Mill\ River}$ is the average recharge of a month in the Mill River either during 2021–2050 or 2051–2080, $R_i$ is the average recharge of a month in the $i$th subbasin (computed in temporal aggregation), and $A_i$ is the area of the $i$th subbasin. This is how average monthly recharges were computed for all five watersheds, for all 12 months for 2021–2050 and 2051–2080 under all the RCP2.6, 4.5, and 8.5 scenarios, and were statistically analyzed to find significant differences across climate normals and RCP scenarios [19,31]. A logical flow diagram broadly presenting the research problems/questions and the methodologies adopted as per worldwide studies and practices, as explained in the above sections, is provided in Figure 3.

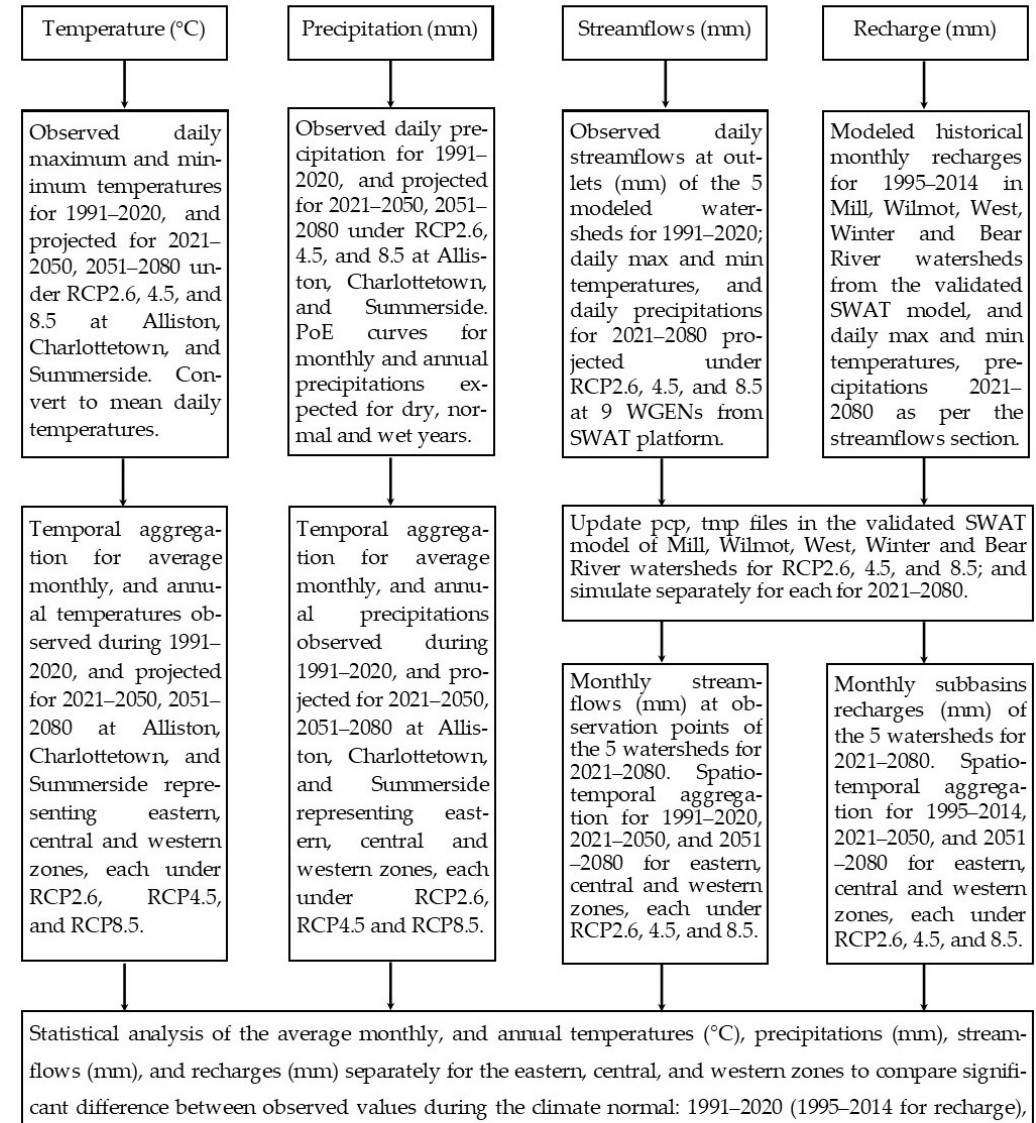

**Figure 3.** Logical flow diagram of the methodologies adopted under the study (RCP: Representative Concentration Pathways).

## 3. Results and Discussion

Hydrological parameters analyzed under the study include temperature, precipitation, streamflow, and groundwater recharge, and results have been described in the same sequence. Those are presented in spatial and temporal disaggregation for better understanding of trends and projections. The spatial divisions include eastern, central, and western PEI, each of which has temporal divisions, i.e., climate normals: 1991–2020, 2021–2050, and 2051–2080. For temperature and precipitation, 1991–2020's information is based on observed data, whereas those of 2021–2050 and 2051–2080 are PCIC projections under the RCP2.6, RCP4.5, and RCP8.5 scenarios. Since streamflows and groundwater recharge has been estimated at the watershed scales, wherein the Mill River and Wilmot River watersheds represent western PEI, the West and Winter River watersheds represent central PEI, and the Bear River watershed represents eastern PEI. For streamflows (mm/year), comparisons have been made for each zone, between observed data of 1991–2020 to those simulated by SWAT during 2021–2050 and 2051–2080 under different

RCP scenarios. For groundwater recharge, baseline values are the modeled values of 1995–2014 and SWAT projections for 2021–2050 and 2051–2080. Pumping-adjusted stream-flows (mm/year) for the Winter and Wilmot River watersheds are also given in the respective zone tables, as detailed in the materials and methods section.

### 3.1. Eastern PEI

Historically, the maximum warming of the island during the last 30 years (1991–2020) took place in the eastern region, wherein the average annual temperature went up from 5.72 °C to 6.86 °C, i.e., a rise of 1.15 °C as compared to that during 1961–1990. Interestingly, the eastern side of the island had undergone a bit of cooling from the 1950s (6.26 °C) to 1970s (5.72 °C), which provides evidence of climate periodicity here [19]. Conversely, significant warming of the region is expected in the future (Figure 4, Table 4); not much difference was diagnosed between the different RCP scenarios during 2021–2050, but it was significant during 2051–2080. Average annual temperature prevailed during 1991–2020, i.e., the 6.86 °C is likely to go up to 8.44–8.62 °C, and to 9.25–11.06 °C during 2021–2050 and 2051–2080, respectively. The results are in accordance with global projections [1,2]. The government of PEI also projects a warming of 1.6 °C by the 2050s [38], and Arnold and Fenech [39] predicted a minimum rise of 2.4 °C on the island by the 2080s.

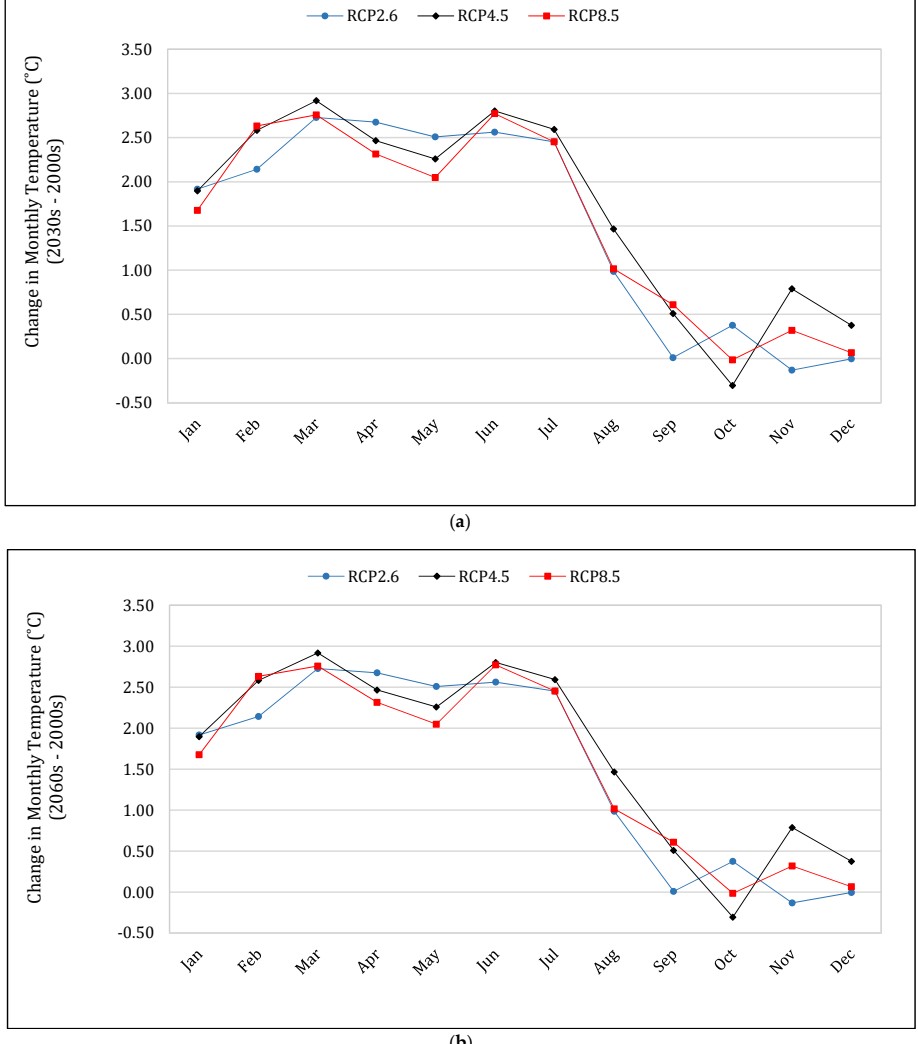

**Figure 4.** Projected change in monthly temperatures during 2021–2050 (**a**), and 2051–2080 (**b**) in the eastern zone (Alliston) compared to those that prevailed during 1991–2020.

**Table 4.** Average temperatures (°C) that prevailed in the eastern zone (Alliston) during the 2000s, and expected under different projections during the 2030s and 2060s.

| Temperature (°C) | 1991–2020 | 2021–2050 | | | 2051–2080 | | |
|---|---|---|---|---|---|---|---|
| | Observed | RCP2.6 | RCP4.5 | RCP8.5 | RCP2.6 | RCP4.5 | RCP8.5 |
| January | −5.83 [c] | −3.92 [b] | −3.94 [b] | −4.16 [b] | −3.22 [ab] | −2.53 [ab] | −1.86 [a] |
| February | −5.79 [c] | −3.65 [b] | −3.21 [b] | −3.61 [b] | −1.05 [a] | −0.65 [a] | 0.39 [a] |
| March | −1.66 [c] | 1.06 [b] | 1.25 [b] | 1.09 [b] | 3.42 [a] | 3.77 [a] | 4.73 [a] |
| April | 3.61 [c] | 6.29 [b] | 6.09 [b] | 5.93 [b] | 8.43 [a] | 8.40 [a] | 9.56 [a] |
| May | 9.83 [d] | 12.34 [c] | 12.09 [c] | 11.88 [c] | 14.23 [b] | 14.55 [b] | 16.16 [a] |
| June | 15.37 [e] | 17.94 [d] | 18.18 [d] | 18.15 [d] | 19.65 [c] | 20.80 [b] | 22.39 [a] |
| July | 19.68 [d] | 22.14 [c] | 22.28 [c] | 22.15 [c] | 22.81 [bc] | 23.47 [b] | 25.75 [a] |
| August | 19.36 [c] | 20.35 [b] | 20.83 [b] | 20.39 [b] | 19.97 [bc] | 20.82 [b] | 22.65 [a] |
| September | 15.32 [bc] | 15.33 [bc] | 15.83 [b] | 15.93 [b] | 14.56 [c] | 15.42 [bc] | 17.01 [a] |
| October | 9.68 [abc] | 10.06 [ab] | 9.38 [bc] | 9.67 [abc] | 8.85 [c] | 9.28 [bc] | 10.57 [a] |
| November | 4.30 [ab] | 4.18 [ab] | 5.09 [a] | 4.62 [ab] | 3.84 [b] | 3.84 [b] | 4.66 [ab] |
| December | −1.51 [b] | −1.52 [b] | −1.14 [ab] | −1.45 [ab] | −1.11 [ab] | −1.28 [ab] | 0.11 [a] |
| **Annual** | **6.86** [e] | **8.44** [d] | **8.62** [cd] | **8.44** [d] | **9.25** [bc] | **9.71** [b] | **11.06** [a] |

Note: Means that do not share a letter are significantly different at α = 0.05 (95% confidence) as per Tukey's method. Bolded values represent average values of all the months to represent average annuals

Maximum warming would take place in late winter and spring, and somewhat in early summer (January–June); the least warming would take place in October, and relatively less in August–December. The warming under all RCP scenarios and durations would be significant during January–July, mostly in August, but not in the rest of the months (September–December). The government of Canada also projects maximum warming to take place during the colder months here [40]. Historically, however, though warming was statistically distributed throughout the year, that in January, March, and June was not significant from the 1970s to 2000s [19], and Garbary [33] also found maximum historical warming here in the months of February, September, and December over the same periods [33]. Hodgekins et al. [41] reported early springs in Atlantic Canada, and the government of Canada also found maximum warming during the winters [21]. The impacts would therefore be moderated colds in winters, earlier springs, and somewhat hotter summers, which would minimally extend to fall and early winter. The island is surrounded by the Gulf of St. Lawrence, which moderates the intensity of both hotness and coldness. Therefore, warming would have an overall beneficial impact on the island's climate, e.g., relatively longer growing seasons, moderated colds, etc. Precipitation in the region is evenly distributed throughout the year, wherein average monthly variations range from 80–130 mm [42], as is evident from Table 5, and it is likely to remain the same with some temporal redistribution. Precipitation changes were found to be insignificant; only September's rainfall would significantly reduce from 101 mm to 64 mm and that too under RCP8.5 during the 2060s. Other than that one, all scenarios, annual, and monthly values present insignificant variations compared to those observed during 1991–2020, though they would be somewhat temporally redistributed.

**Table 5.** Average precipitations (mm) observed in the eastern zone (Alliston) during the 2000s and expected in normal years (PoE = 50%) under different projections during the 2030s and 2060s.

| Precipitation (mm) | 1991–2020 | 2021–2050 | | | 2051–2080 | | |
|---|---|---|---|---|---|---|---|
| | Observed | RCP2.6 | RCP4.5 | RCP8.5 | RCP2.6 | RCP4.5 | RCP8.5 |
| January | 102 [a] | 101 [a] | 110 [a] | 99 [a] | 106 [a] | 99 [a] | 116 [a] |
| February | 87 [a] | 86 [a] | 97 [a] | 98 [a] | 83 [a] | 101 [a] | 109 [a] |
| March | 92 [a] | 98 [a] | 90 [a] | 104 [a] | 99 [a] | 98 [a] | 104 [a] |
| April | 86 [a] | 89 [a] | 99 [a] | 96 [a] | 90 [a] | 89 [a] | 100 [a] |
| May | 81 [a] | 80 [a] | 91 [a] | 97 [a] | 92 [a] | 81 [a] | 80 [a] |
| June | 84 [a] | 95 [a] | 96 [a] | 86 [a] | 82 [a] | 85 [a] | 84 [a] |
| July | 81 [a] | 97 [a] | 76 [a] | 91 [a] | 90 [a] | 101 [a] | 71 [a] |
| August | 95 [a] | 90 [a] | 91 [a] | 102 [a] | 86 [a] | 98 [a] | 85 [a] |
| September | 101 [a] | 85 [ab] | 94 [ab] | 90 [ab] | 85 [ab] | 80 [ab] | **64 [b]** |
| October | 122 [a] | 101 [a] | 119 [a] | 86 [a] | 106 [a] | 101 [a] | 112 [a] |
| November | 131 [a] | 125 [a] | 114 [a] | 129 [a] | 111 [a] | 129 [a] | 132 [a] |
| December | 120 [a] | 113 [a] | 115 [a] | 120 [a] | 123 [a] | 118 [a] | 119 [a] |
| **Annual** | **1182 [a]** | **1160 [a]** | **1192 [a]** | **1198 [a]** | **1153 [a]** | **1180 [a]** | **1176 [a]** |

Note: Means that do not share a letter are significantly different at $\alpha$ = 0.05 (95% confidence) as per Tukey's method. Bolded values represent total of all the months to represent average annuals.

The RCP4.5 scenario presents an insignificant increase in precipitation during July–December and vice-versa for the rest of the months during the 2030s, whereas during the 2060s, precipitation is likely to decrease during September to January and increase in the rest of the months. Bhatti et al. [19] found a 6% increase in annual precipitation here during 1991–2020 (1182 mm) compared to that which prevailed during 1961–1990 (1116 mm); particularly, snowfall in the month of March did show a significant increase (+20%). Moreover, precipitation has been continuously increasing on eastern PEI since the 1930s (837 mm) [19]. However, no further increase in the precipitation quantum is expected, as it might have already achieved its maximum. The results are in accordance with DesJarlais et al. [43], who projected inappreciable changes in precipitation in Southeastern Canada by 2050 [31]. The government of PEI also projects minimal changes in prospective precipitation up till the 2080s [38]. Nevertheless, rainfall and snowfall intensity might increase under the same overall quantum, which, however, needs monitoring. It was reported that the annual maximum daily precipitation in Atlantic Canada had increased during 1990–2000 [44], which, as well as a recent study, supports a rise in rainfall intensity [19].

Historically (1991–2020), the precipitations expected in dry and wet years were 1010 mm/year and 1340 mm/year, respectively. However, further minimal variations are expected, and dry years would range from 1020–1050 mm/year, and wet years from 1280–1350 mm/year. It was reported that uncertainty in precipitation has narrowed here in the last 30 years [19], and would further narrow slightly to ~300 mm (Figure 5). Despite minimal further variations in the annual precipitations in dry and wet years, there would be significant redistribution on monthly scales. The variations are relatively small in dry years and large in wet years. Particularly, the RCP8.5 scenario and those during 2051–2080 depict sharp variations in expected monthly precipitation. From the water management standpoint, a reduction of the gap of precipitation in dry and wet years at the annual scale is good, but its monthly redistribution can pose challenges and therefore needs due consideration in regulations. Doria and Madramootoo [26] also found irrigation requirements to increase ~2 times during dry years compared to normal years in southern Quebec [26].

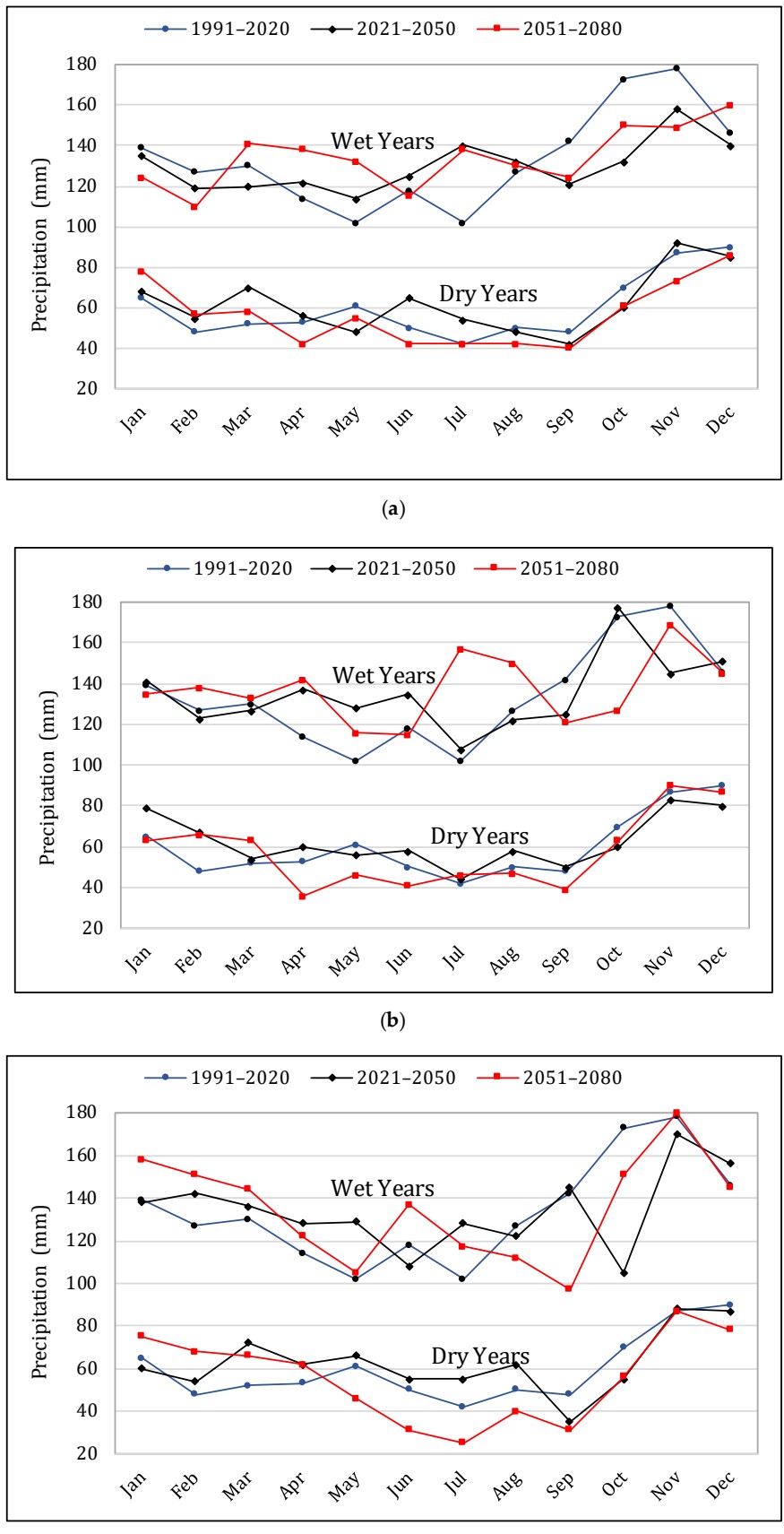

**Figure 5.** Expected monthly precipitations during wet (PoE = 20%), and dry years (PoE = 80%) under RCP2.6 (**a**), RCP4.5 (**b**), and RCP8.5 (**c**) scenarios at in the eastern zone (Alliston).

The validated SWAT model was simulated under different RCP scenarios by inputting prospective precipitation and temperatures data, ascertained from PCIC for 2021–2080. The simulated mean monthly streamflows ($m^3$/s) were extracted from the respective subbasins of monitoring points, i.e., subbasin 8 (Figure 2) for the Bear River watershed, representing the eastern zone. Streamflows were converted into mm to better grasp the results, and statistically analyzed to find significant variations across climate normals, as shown in Table 6.

**Table 6.** Average streamflows (mm) observed in the eastern zone (at St. Margaret's of the Bear River watershed) during the 2000s and modeled future projections during the 2030s and 2060s.

| Streamflows (mm) | 1991–2020 | 2021–2050 | | | 2051–2080 | | |
|---|---|---|---|---|---|---|---|
| | Observed | RCP2.6 | RCP4.5 | RCP8.5 | RCP2.6 | RCP4.5 | RCP8.5 |
| January | 67 [b] | **90** [a] | **92** [a] | 87 [ab] | **92** [a] | 88 [ab] | **97** [a] |
| February | 44 [b] | **79** [a] | **91** [a] | **78** [a] | 87 [a] | 95 [a] | 88 [a] |
| March | 63 [b] | 78 [ab] | 80 [ab] | 93 [a] | 77 [ab] | 77 [ab] | 89 [a] |
| April | 129 [a] | **66** [b] | **74** [b] | **76** [b] | 71 [b] | 73 [b] | 78 [b] |
| May | 81 [a] | **66** [b] | 69 [ab] | 73 [ab] | 63 [b] | 62 [b] | 67 [ab] |
| June | 42 [b] | 47 [ab] | 54 [ab] | 58 [a] | 47 [ab] | 46 [ab] | 47 [ab] |
| July | 26 [b] | 38 [ab] | 40 [a] | 41 [a] | 34 [ab] | 34 [ab] | 29 [ab] |
| August | 24 [ab] | 31 [a] | 26 [ab] | 32 [a] | 22 [ab] | 27 [ab] | 17 [b] |
| September | 31 [a] | 23 [ab] | 18 [ab] | 22 [ab] | 16 [b] | 19 [ab] | 9 [b] |
| October | 42 [a] | 27 [ab] | 30 [ab] | **24** [b] | 23 [b] | 23 [b] | 14 [b] |
| November | 64 [a] | 44 [ab] | **43** [b] | **37** [b] | 38 [b] | 41 [b] | 33 [b] |
| December | 79 [a] | 74 [a] | 71 [a] | 76 [a] | 75 [a] | 78 [a] | 69 [a] |
| **Annual** | **692** [a] | **663** [a] | **688** [a] | **697** [a] | **645** [a] | **663** [a] | **637** [a] |

Note: Means that do not share a letter are significantly different at $\alpha$ = 0.05 (95% confidence) as per Tukey's method. Bolded values highlight significantly different ones in the text, the annual are bolded because they represent sum total of all the months.

The Atlantic region is the area with the highest streamflow-generating potential, ranging from 600–1200 mm/year, against the Canadian average of 350 mm/year [45,46]. Accordingly, the eastern region generated ~700 mm/year during 1991–2020, of which 17% was direct runoff. This is because the watershed is almost entirely forested, mildly sloped (0–5%), and has highly infiltrating soils. No significant change in that is likely; nevertheless, annual streamflows could potentially reduce by 8% under RCP8.5 during 2051–2080. It indicates that eastern PEI is somewhat resilient to climate change, as several other studies revealed that runoff in different regions around the world has already and is expected to further change significantly [11]. The temporal distribution of flows would change significantly, wherein September would be the month of least flows. This is in accordance with the likely precipitation pattern, and streamflows would be more evenly distributed in a year. At the worldwide scale, spring flows increased and summer flows reduced because of warming, and more intense rainfall events occurred [41]. Accordingly, high streamflows here in April–May would be significantly reduced because of higher flows in January, February, and March. This correlates with the expected warming trend also leading to early snowmelt and an earlier contribution to streamflows, and thus justifies the streamflow trend. The same trend has been projected that most North American rivers are to get streamflow peaks 30–40 days earlier [12]. December would be the least affected month; low flows are likely from September–November.

Most PEI soils have high infiltration and low runoff potential [17,47]. Compounded by high hydraulic conductivity and relatively flat topography, PEI has one of the highest recharge rates [22]. However, the largest part of that joins streamflows [20,22], and that is why a high overlap in the renewable water components is found. Groundwater recharges in the eastern zone were found the highest in PEI at about 600 mm/year during 1995–2014, as shown in Table 7 [20].

**Table 7.** Average recharge (mm) in the eastern zone (Bear River watershed) during 1995–2014 and modeled future projections during the 2030s and 2060s.

| Recharge (mm) | 1995–2014 | 2021–2050 | | | 2051–2080 | | |
|---|---|---|---|---|---|---|---|
| | Historical | RCP2.6 | RCP4.5 | RCP8.5 | RCP2.6 | RCP4.5 | RCP8.5 |
| January | 39 ab | 46 b | 48 ab | 44 b | 53 ab | 54 ab | 63 a |
| February | 28 bcd | 32 d | 33 cd | 28 d | 49 ab | 49 abc | **60 a** |
| March | 38 c | 58 bc | 61 bc | 60 bc | 76 ab | **79 a** | **85 a** |
| April | 64 a | 70 a | 73 a | 82 a | 63 a | 64 a | 68 a |
| May | 85 a | **55 b** | **57 b** | **60 b** | **50 b** | **48 b** | **54 b** |
| June | 69 a | **36 bc** | **42 bc** | **46 b** | **36 bc** | **33 c** | **33 c** |
| July | 58 a | **31 b** | **29 b** | **29 b** | **24 b** | **26 b** | **21 b** |
| August | 46 a | **28 b** | **20 bc** | **24 bc** | **19 bc** | **22 bc** | **12 c** |
| September | 38 a | 21 ab | **17 b** | 21 ab | **17 b** | **18 b** | **9 b** |
| October | 39 a | 27 ab | 32 a | 26 ab | 27 ab | 24 ab | **13 b** |
| November | 42 a | 52 a | 54 a | 47 a | 47 a | 50 a | 42 a |
| December | 51 a | 68 a | 70 a | 69 a | 65 a | 72 a | 70 a |
| **Annual** | **597 a** | **524 a** | **536 a** | **536 a** | **526 a** | **539 a** | **530 a** |

Note: Means that do not share a letter are significantly different at α = 0.05 (95% confidence) as per Tukey's method. Bolded values highlight significantly different ones in the text, the annual are bolded because they represent sum total of all the months.

No significant change is likely in the prospective annual recharges in the Bear River watershed, though it could reduce up to 12% with temporal redistribution (Table 7). This is so because, as per the model results, average annual ET during 1995–2014 was 474 mm/year, which would increase by 3%, 4%, and 7% under RCP2.6, 4.5, and 8.5, respectively during 2021–2080. More recharge is likely in the winter months, and significantly less during summer because of higher temperatures, ET, and near constant precipitation. This is in accordance with a recent Canadian study, wherein increased recharge was reported during late winter and spring because of reduced frost and more infiltration, and decreased recharge was reported in summer because of more intense rainfall events [15]. Nevertheless, groundwater pumping in the watershed is negligible (1 mm/year); therefore, non-climatic factors are less likely to further affect the water balance. It can therefore be concluded that the overall water balance of the eastern zone would be negatively affected up to ~100 mm/year mainly because of a rise in ET losses. Accordingly, warming-induced rises in ET have also reduced water availability in other parts of the world [11]. Despite lesser variations at the annual scale, temporal redistribution of streamflows and recharges may be duly considered for better water management.

### 3.2. Central PEI

Warming observed from the 1970s to 2000s was relatively gentler in central PEI, during which the average annual temperature went up by +0.70 °C, compared to +1.14 °C in the eastern zone [19]. Statistically significant warming is expected in the next 30 to 60 years, as evident from Figure 6 and Table 8.

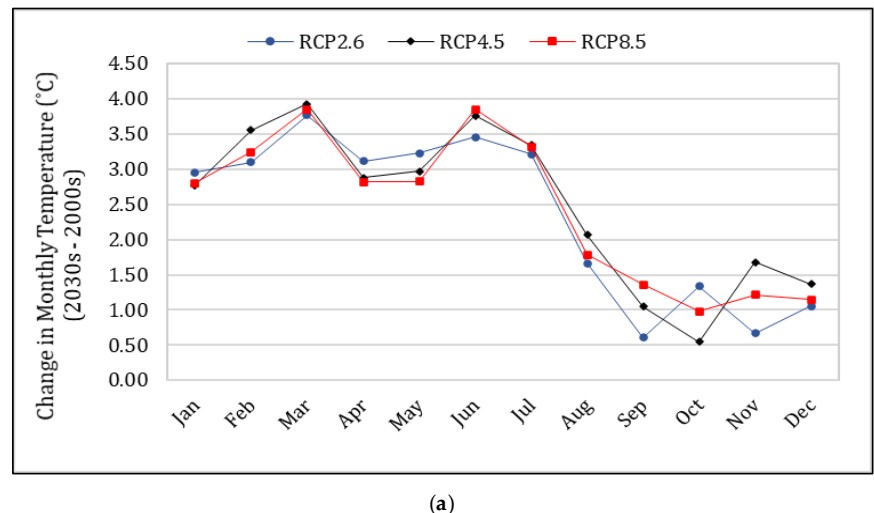

(**a**)

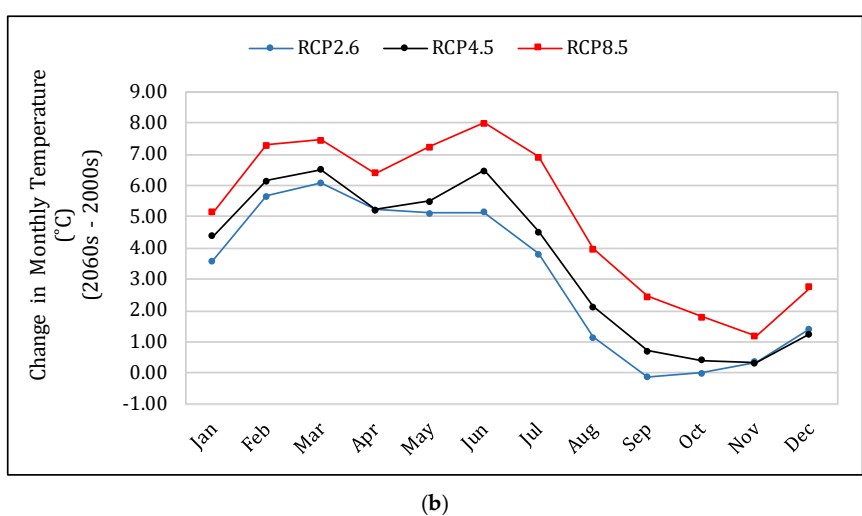

(**b**)

**Figure 6.** Projected change in monthly temperatures during 2021–2050 (**a**) and 2051–2080 (**b**) in the central zone (Charlottetown) compared to those that prevailed during 1991–2020.

**Table 8.** Average temperatures (°C) that prevailed in the central zone (Charlottetown) during the 2000s and expected under different projections during the 2030s and 2060s.

| Temperature (°C) | 1991–2020 | 2021–2050 | | | 2051–2080 | | |
|---|---|---|---|---|---|---|---|
| | Observed | RCP2.6 | RCP4.5 | RCP8.5 | RCP2.6 | RCP4.5 | RCP8.5 |
| January | −7.20 [c] | −4.25 [b] | −4.43 [b] | −4.40 [b] | −3.60 [ab] | −2.83 [ab] | −2.06 [a] |
| February | −7.09 [c] | −4.00 [b] | −3.54 [b] | −3.85 [b] | −1.42 [a] | −0.94 [a] | 0.21 [a] |
| March | −2.88 [d] | 0.89 [c] | 1.05 [c] | 0.97 [c] | 3.21 [b] | 3.63 [ab] | 4.58 [a] |
| April | 3.09 [d] | 6.22 [c] | 5.98 [c] | 5.92 [c] | 8.33 [b] | 8.33 [b] | 9.49 [a] |
| May | 9.14 [d] | 12.37 [c] | 12.11 [c] | 11.97 [c] | 14.25 [b] | 14.65 [b] | 16.40 [a] |
| June | 14.58 [e] | 18.04 [d] | 18.34 [d] | 18.43 [d] | 19.74 [c] | 21.05 [b] | 22.60 [a] |
| July | 18.98 [d] | 22.20 [c] | 22.33 [c] | 22.31 [c] | 22.79 [bc] | 23.50 [b] | 25.92 [a] |
| August | 18.74 [d] | 20.4 [bc] | 20.81 [b] | 20.54 [bc] | 19.90 [c] | 20.88 [b] | 22.73 [a] |
| September | 14.63 [c] | 15.25 [bc] | 15.70 [b] | 16.00 [b] | 14.51 [c] | 15.36 [bc] | 17.08 [a] |
| October | 8.64 [c] | 9.98 [ab] | 9.19 [bc] | 9.62 [ab] | 8.63 [c] | 9.06 [bc] | 10.45 [a] |
| November | 3.12 [c] | 3.80 [abc] | 4.82 [a] | 4.34 [ab] | 3.47 [bc] | 3.46 [bc] | 4.31 [abc] |
| December | −2.92 [b] | −1.86 [b] | −1.55 [ab] | −1.77 [b] | −1.50 [ab] | −1.68 [ab] | −0.18 [a] |
| **Annual** | **5.90** [d] | **8.32** [c] | **8.46** [c] | **8.40** [c] | **9.07** [b] | **9.59** [b] | **11.01** [a] |

Note: Means that do not share a letter are significantly different at α = 0.05 (95% confidence) as per Tukey's method. The bolded values represent annual average of all the months.

The average annual temperature would increase from 5.90 °C to 8.46 °C during 2021–2050, and to 9.59 °C during 2051–2080 under the moderate emissions scenario (RCP4.5), and similarly for other scenarios (Table 8). The results resemble the previous forecast by Arnold and Fenech [39]. The impacts during 2051–2080 present a clear picture of the maximum warming under RCP8.5, followed by RCP4.5, and RCP2.6, respectively. Similar to what was observed for eastern PEI, warming is mostly concentrated in late winter, spring, and early summer, and RCP8.5 during the 2060s presents maximum warming. Accordingly, Jiang et al. [28] reported the maximum rise in temperature during the winter months for Central Canada. The intra-annual distribution of warming for the months January–August is also statistically significant for all the scenarios compared to the 2000s. Historically, rises in the mean monthly minimum temperature were more than the mean monthly maximum, and indicated moderated colds [19]. Garbary [33] found average monthly temperatures to increase from 0.3 to 1.6 °C at Charlottetown more during 1991–2014 than 1961–1990 with the maximum increase in December [33]. The trend would continue to moderate colds, and to cause early springs and somewhat hotter summers. The results agree with the IPCC projections [1,2], DesJarlais et al. [43], and the government of PEI [38].

In contrast to the temperature trends, precipitation is likely to increase statistically insignificantly. It is evident from Table 9 that the precipitation that prevailed during 1991–2020 would increase from 1141 mm/year up to 1192 mm/year during 2051–2080 under RCP4.5. However, no significant change is likely. Arnold and Fenech [39] forecast a 6% decline in precipitation in Charlottetown during 2011–2040, which contradicts the findings. Perhaps the historic decline that occurred during the last decade of the climate normal, 1991–2020, has been dominantly reflected in the forecast. The government of PEI also projects minimal variations in the annual precipitations by 2080 with temporal redistribution [38]. Temporal distribution of precipitation also does not show any significant increase, except in the month of February, wherein the snowfall may significantly increase from 86 cm to 115 cm under RCP8.5 during 2051–2080. Inter-relating with historic trends will reveal that the projected increase of precipitation is similar to what happened in the past, as precipitation non-significantly increased from 1104 mm/year in the 1950s to 1141 mm/year in the 2000s. Rainfall intensity during 2007–2016 was reported to increase by 8–32% as compared to in the 1970s [19]. Near-constant rainfall with more intensity would negatively affect water availability but would be counterbalanced by more snowmelt during spring.

**Table 9.** Average precipitations (mm) observed in the central zone (Charlottetown) during the 2000s and expected under different projections during the 2030s and 2060s.

| Precipitation (mm) | 1991–2020 | 2021–2050 | | | 2051–2080 | | |
|---|---|---|---|---|---|---|---|
| | Observed | RCP2.6 | RCP4.5 | RCP8.5 | RCP2.6 | RCP4.5 | RCP8.5 |
| January | 100 [a] | 103 [a] | 112 [a] | 102 [a] | 107 [a] | 98 [a] | 121 [a] |
| February | 86 [b] | 87 [ab] | 101 [ab] | 97 [ab] | 87 [ab] | 105 [ab] | 115 [a] |
| March | 84 [a] | 100 [a] | 93 [a] | 106 [a] | 101 [a] | 95 [a] | 106 [a] |
| April | 80 [a] | 88 [a] | 102 [a] | 100 [a] | 91 [a] | 97 [a] | 104 [a] |
| May | 79 [a] | 86 [a] | 96 [a] | 106 [a] | 98 [a] | 86 [a] | 83 [a] |
| June | 88 [a] | 100 [a] | 95 [a] | 86 [a] | 86 [a] | 89 [a] | 87 [a] |
| July | 79 [ab] | 104 [ab] | 77 [ab] | 93 [ab] | 92 [ab] | 105 [a] | 71 [b] |
| August | 96 [a] | 86 [a] | 90 [a] | 93 [a] | 80 [a] | 94 [a] | 75 [a] |
| September | 94 [a] | 82 [a] | 87 [a] | 84 [a] | 80 [a] | 75 [a] | 61 [a] |
| October | 118 [a] | 101 [a] | 119 [a] | 86 [a] | 102 [a] | 95 [a] | 110 [a] |
| November | 112 [a] | 120 [a] | 108 [a] | 122 [a] | 109 [a] | 128 [a] | 130 [a] |
| December | 125 [a] | 114 [a] | 118 [a] | 122 [a] | 122 [a] | 125 [a] | 124 [a] |
| **Annual** | **1141 [a]** | **1171 [a]** | **1198 [a]** | **1197 [a]** | **1155 [a]** | **1192 [a]** | **1187 [a]** |

Note: Means that do not share a letter are significantly different at $\alpha$ = 0.05 (95% confidence) as per Tukey's method. Bolded values represent total of all the months to represent average annuals.

Changes in precipitation expectancy during the wet and dry years are provided in Figure 7. Historically, precipitation expectancy during dry and wet years ranged from 960–1320 mm/year. This presents interannual variations of up to 360 mm/year; dry years were drier than in eastern PEI (1010 mm/year). The gap is, however, likely to be slightly narrowed to between 1000–1325 mm/year. This indicates that dry years would become somewhat wetter, and the change would be more pronounced during 2021–2050. However, slightly higher precipitation during dry years is not likely to increase net water availability because of the risen temperatures and increased ET thereof [48]. As per the model's results, average annual ET during 1995–2014 in the zone was 464 mm/year, which would increase by 7%, 8%, and 11% under RCP2.6, 4.5, and 8.5, respectively, during 2021–2080. A scan of the monthly variations likely under different scenarios and durations also reveals that the range of variations is higher in wet years. Accordingly, frequent occurrence of more wet and dry years have been forecast for Southeastern Canada [26].

The observed and expected streamflows (mm) in the central zone are provided in Table 10. The simulated mean monthly streamflows ($m^3/s$) were extracted from the respective subwatersheds of the monitoring points, i.e., 53 and 25 (Figure 2) for the West and Winter River watersheds, respectively. Weighted average streamflows of the central zone (Figure 2) were converted into mm to better grasp the results, as shown in Table 10.

Unlike in the eastern zone, streamflows in the central zone can significantly reduce from 737 mm/year to 621 mm/year (−16%) under RCP8.5 during the 2060s, as evident from Table 10. The likely reductions during 2021–2050 are not statistically significant. Direct runoff in the zone was ~200 mm/year (28% of streamflows), while the rest were baseflow contributions. The results agree with the global projections, which mostly envision decreased streamflows and water availability caused by warming and more ET thereof [11,49,50]. The temporal distribution of the streamflows would be more even, with higher flows in January, February, and March and relatively lesser ones in April–May. Comparatively higher flows are likely in June–July, and lower in August–December. Attenuation of streamflows can be directly correlated with the warming trend, causing early snowmelt and streamflows thereof during winter and spring, and somewhat lesser ones in summer, as reported before [41]. Overall streamflow availability might go down but would be more attenuated.

Streamflows in the Winter River watershed are likely to reduce because of the combined effects of warming and pumping, as there is negligible pumping in the West River watershed. Pumping-adjusted streamflows of the Winter River watershed are given at the bottom of Table 10. As per the previous findings and per the adopted methodology, every 1 mm/year of pumping will equally reduce streamflows. Streamflows can significantly go down from the present 565 mm/year (1991–2020) to 429 mm/year during 2051–2080, if the present pumping continues in the future, i.e., 191 mm/year. This would be a reduction of about 24% caused by both climatic factors and pumping. Streamflows are particularly vulnerable during the summer months when the maximum pumping of the year takes place. Therefore, the issuance of more pumping permits for the Winter River watershed should only be done after careful hydrogeological scrutiny [22]. Annual and seasonal pumping quotas and water meters should be allocated for sustainable water management in the Winter River watershed.

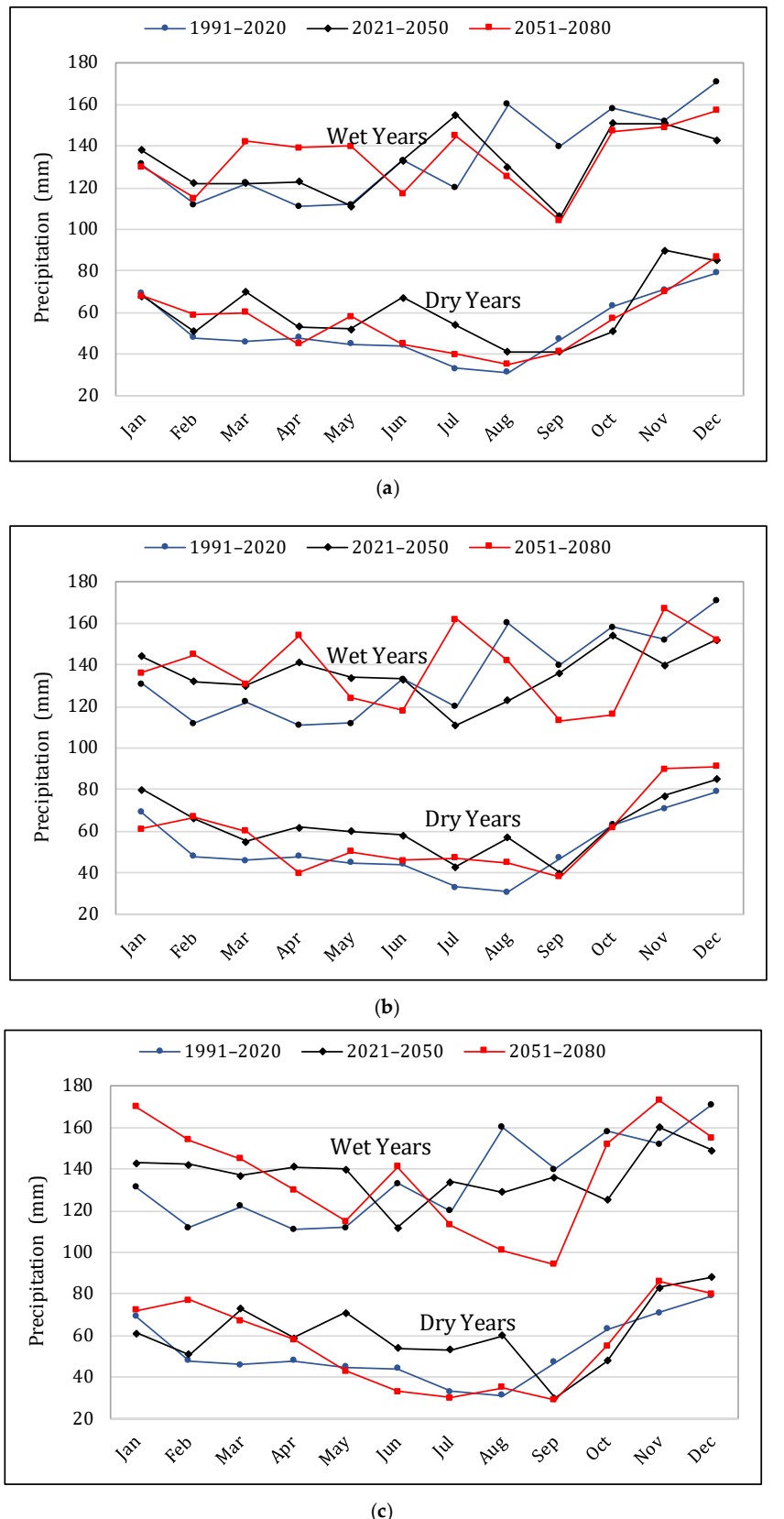

**Figure 7.** Expected monthly precipitations during wet (PoE = 20%) and dry years (PoE = 80%) under the RCP2.6 (**a**), RCP4.5 (**b**), and RCP8.5 (**c**) scenarios in the central zone (Charlottetown).

**Table 10.** Average streamflows (mm) observed in the central zone (at Riverdale and Suffolk of the West and Winter River watersheds, respectively) during the 2000s and modeled future projections during the 2030s and 2060s.

| Streamflows (mm) | 1991–2020 | 2021–2050 | | | 2051–2080 | | |
|---|---|---|---|---|---|---|---|
| | Observed | RCP2.6 | RCP4.5 | RCP8.5 | RCP2.6 | RCP4.5 | RCP8.5 |
| January | 69 a | 81 a | 80 a | 74 a | 81 a | 78 a | 86 a |
| February | 48 c | 72 abc | **86 ab** | 70 bc | **87 ab** | **95 a** | **89 ab** |
| March | 80 a | 80 a | 88 a | 97 a | 79 a | 80 a | 91 a |
| April | 138 a | **69 b** | **82 b** | **85 b** | **72 b** | **77 b** | **81 b** |
| May | 92 a | **67 b** | 72 ab | 77 ab | **67 b** | **66 b** | 69 ab |
| June | 47 b | 52 ab | 57 ab | **61 a** | 50 ab | 50 ab | 49 ab |
| July | 33 ab | 46 a | 41 ab | 45 a | 36 ab | 39 ab | 29 b |
| August | 27 a | 32 a | 24 ab | 29 a | 23 ab | 25 ab | **13 b** |
| September | 28 a | 22 ab | 16 abc | 19 ab | **16 bc** | **15 bc** | **7 c** |
| October | 39 a | 24 abc | 31 ab | **19 bc** | **21 bc** | **18 bc** | **15 c** |
| November | 59 a | 39 b | 38 b | 33 b | 34 b | 35 b | 29 b |
| December | 77 a | 68 a | 65 a | 65 a | 67 a | 71 a | 63 a |
| **Annual** | **737 a** | **652 ab** | **680 ab** | **674 ab** | **633 b** | **649 ab** | **621 b** |
| **Winter River** | **565** | **(453)** | **(481)** | **(479)** | **(434)** | **(448)** | **(429)** |

Note: Means that do not share a letter are significantly different at α = 0.05 (95% confidence) as per Tukey's method.
Note: bracketed values are pumping adjusted as detailed in Section 2.3. Bolded values highlight significantly different ones in the text, the annual are bolded because they represent sum total of all the months.

Groundwater recharge in the central zone is not likely to significantly decrease, though it might go down by 9% from the present 446 mm/year to 405 mm/year during 2051–2080 (Table 11). The moderate emission scenario (RCP4.5) projects almost no change in the present recharge during 2051–2080. Temporal redistribution would occur with increased recharge during winter and spring (January–May), and mostly significant reduction would occur during the summers (June–October), whereas that occurring during fall (November–December) would almost remain the same. These results are in accordance with the comprehensive Canadian review indicating more recharges in winter and spring, and less in summer [15]. The likely reduction of 40 mm/year (9%) in recharges and up to 16% in streamflows is due to warming and increased ET thereof. This is in accordance with another study of the Soan River Basin in Pakistan, where PET, land-use, and precipitation changes have significantly reduced streamflows [11]. In the Winter River watershed, recharge is likely to significantly go down by ~100 mm/year, with almost 20% of the reduction caused by warming, which in turn would reduce the baseflow contribution. A 7–11% increase in ET is partially responsible for the reduction. The inter-scenario differences are small, which indicates that warming beyond certain limits would not affect recharges. Li [22] projected an increase of 7–14% in PEI recharges under different RCP scenarios in the next 30–60 years, caused by more snowmelt in winter and spring. However, the study projects the somewhat reduced recharges become justified as per the previous findings, which state, "thawing permafrost, increases soil infiltration, and groundwater recharge thereof, can have a net positive effect on recharge at places, where proportionate increase in precipitation was more than evapotranspiration" [40,51,52].

**Table 11.** Average recharge (mm) in the central zone (West, and Winter River watersheds) during 1995–2014, and modeled future projections during the 2030s and 2060s.

| Recharge (mm) | 1995–2014 | 2021–2050 | | | 2051–2080 | | |
|---|---|---|---|---|---|---|---|
| | Historical | RCP2.6 | RCP4.5 | RCP8.5 | RCP2.6 | RCP4.5 | RCP8.5 |
| January | 29 ab | 34 b | 38 ab | 32 b | 37 ab | 40 ab | 48 a |
| February | 21 c | 25 c | 28 bc | 23 c | **41 ab** | **40 ab** | **53 a** |
| March | 29 d | **51 c** | 50 cd | 47 cd | **66 b** | **85 a** | **88 a** |
| April | 51 ab | 63 ab | 65 ab | 74 a | 55 b | 58 ab | 61 ab |
| May | 66 a | 49 ab | 51 ab | 54 ab | **44 b** | **44 b** | 48 ab |
| June | 53 a | **33 bc** | **35 bc** | 41 ab | **32 bc** | **29 c** | **28 c** |
| July | 44 a | 28 b | 24 bc | 25 bc | 20 bc | 21 bc | 17 c |
| August | 34 a | **22 b** | **13 c** | **15 bc** | **14 bc** | **15 bc** | **7 c** |
| September | 27 a | **13 b** | **15 bc** | **11 bc** | **10 bc** | **10 bc** | **4 c** |
| October | 26 a | 15 ab | 18 a | 12 ab | 15 ab | 11 ab | **6 b** |
| November | 30 a | 32 a | 35 a | 27 a | 28 a | 28 a | 21 a |
| December | 36 b | 47 ab | 47 b | 47 ab | 43 b | 60 a | 43 b |
| **Annual** | **446 a** | **412 a** | **419 a** | **408 a** | **405 a** | **441 a** | **424 a** |
| **Winter River** | **504** | **404** | **413** | **412** | **403** | **413** | **422** |

Note: Means that do not share a letter are significantly different at $\alpha = 0.05$ (95% confidence) as per Tukey's method. Bolded values highlight significantly different ones in the text, the annual are bolded because they represent sum total of all the months.

### 3.3. Western PEI

Temperature and precipitation analysis of the western zone has been based on historical and projected data at Summerside, whereas streamflow and groundwater recharge analysis has been based on the weighted averages of the Mill and Wilmot River watersheds (Figure 1), as well as individualistically for the Wilmot River with pumping adjustments.

A statistically significant warming of the western zone is expected in the next 60 years (Figure 8, Table 12). Average annual temperature would increase from 6.22 °C to 8.37 °C, and to 9.54 °C under RCP4.5 during the 2030s and 2060s, respectively. Historically, the average annual temperature during the 1970s was 5.47 °C, which also significantly rose to 6.22 °C [19]. This indicates a continuation of the existing trend of warming under all scenarios, with different severities. Previous studies testify to the warming projections, such as those by Arnold and Fenech [39], the government of PEI [38], and the government of Canada [40].

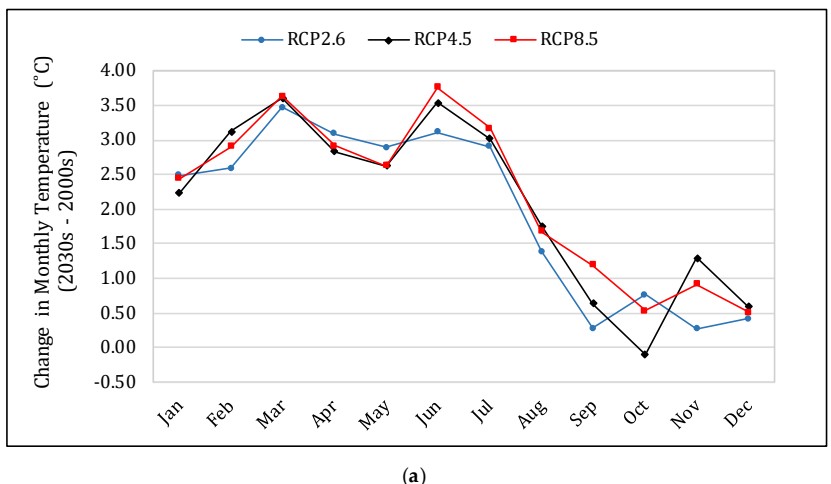

(**a**)

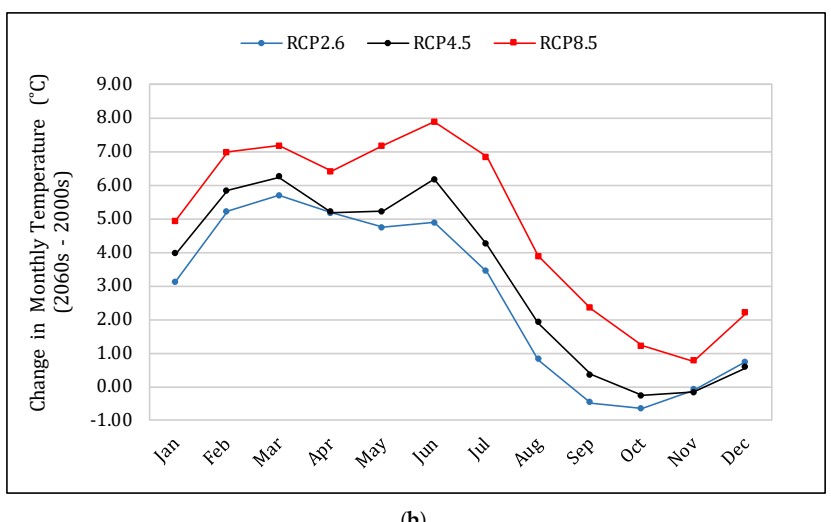

(**b**)

**Figure 8.** Projected change in monthly temperatures during 2021–2050 (**a**) and 2051–2080 (**b**) in the western zone (Summerside), compared to those that prevailed during 1991–2020.

**Table 12.** Average temperatures (°C) that prevailed in the western zone (Summerside) during the 2000s and expected under different projections during the 2030s and 2060s.

| Temperature (°C) | 1991–2020 | 2021–2050 | | | 2051–2080 | | |
|---|---|---|---|---|---|---|---|
| | **Observed** | **RCP2.6** | **RCP4.5** | **RCP8.5** | **RCP2.6** | **RCP4.5** | **RCP8.5** |
| January | −6.96 ᵈ | −4.48 ᵇᶜ | −4.74 ᶜ | −4.53 ᵇᶜ | −3.85 ᵇᶜ | −3.03 ᵃᵇ | −2.05 ᵃ |
| February | −6.72 ᵈ | −4.14 ᶜ | −3.61 ᶜ | −3.83 ᶜ | −1.53 ᵇ | −0.89 ᵃᵇ | 0.25 ᵃ |
| March | −2.57 ᵈ | 0.90 ᶜ | 1.03 ᶜ | 1.06 ᶜ | 3.12 ᵇ | 3.66 ᵃᵇ | 4.60 ᵃ |
| April | 3.17 ᵈ | 6.26 ᶜ | 6.01 ᶜ | 6.09 ᶜ | 8.33 ᵇ | 8.36 ᵇ | 9.58 ᵃ |
| May | 9.59 ᵈ | 12.49 ᶜ | 12.22 ᶜ | 12.22 ᶜ | 14.32 ᵇ | 14.80 ᵇ | 16.76 ᵃ |
| June | 15.06 ᵉ | 18.18 ᵈ | 18.60 ᵈ | 18.83 ᵈ | 19.94 ᶜ | 21.24 ᵇ | 22.95 ᵃ |
| July | 19.41 ᵈ | 22.32 ᶜ | 22.43 ᶜ | 22.58 ᶜ | 22.84 ᶜ | 23.65 ᵇ | 26.26 ᵃ |
| August | 19.06 ᵈ | 20.45 ᵇᶜ | 20.82 ᵇ | 20.74 ᵇ | 19.86 ᶜᵈ | 20.97 ᵇ | 22.94 ᵃ |
| September | 14.89 ᶜᵈ | 15.18ᵇᶜᵈ | 15.53 ᵇᶜ | 16.09 ᵇ | 14.41 ᵈ | 15.26ᵇᶜᵈ | 17.22 ᵃ |
| October | 9.10 ᵇᶜᵈ | 9.87 ᵃᵇ | 9.01 ᵇᶜᵈ | 9.63 ᵃᵇᶜ | 8.45 ᵈ | 8.85 ᶜᵈ | 10.31 ᵃ |
| November | 3.20 ᵇ | 3.47 ᵃᵇ | 4.50 ᵃ | 4.12 ᵃᵇ | 3.10 ᵇ | 3.02 ᵇ | 3.95 ᵃᵇ |
| December | −2.57 ᵇ | −2.16 ᵇ | −1.99 ᵇ | −2.06 ᵇ | −1.84 ᵃᵇ | −2.02 ᵇ | −0.38 ᵃ |
| **Annual** | **6.22 ᵉ** | **8.26 ᵈ** | **8.37 ᵈ** | **8.47 ᶜᵈ** | **8.98 ᶜ** | **9.54 ᵇ** | **11.09 ᵃ** |

Note: Means that do not share a letter are significantly different at α = 0.05 (95% confidence) as per Tukey's method. The bolded values represent annual average of all the months and need be differentiated.

Like in the eastern zone, the central zone's inter-scenario difference is visible during the 2060s, whereas that during the 2030s is small. The warming trend is non-uniformly distributed over a year, as the months January–July show significant warming, and those of August–December lesser warming. In contrast, from the 1970s to 2000s, warming was uniformly distributed throughout a year, and the months of August–October exhibited significant warming [19], whereas Garbary [33] indicated that mean monthly temperatures would decrease in June by −0.2 °C and increase by 1–1.9 °C from October–January in western PEI during 1991–2014, unlike in the 1970s [33]. A rise in temperatures would increase ET [48], as per the model results of average annual ET during 1995–2014 of 460 mm/year, which is likely to increase by 10%, 12%, and 15% under RCP2.6, 4.5, and 8.5, respectively, during 2021–2080.

The precipitation trends in the western zone are quite interesting, as the average annual precipitation is likely to significantly increase from 978 mm/year up to 1157 mm/year (Table 13). Particularly, snowfall during January and February, and rainfall in July would significantly increase. At an overall scale, the government of PEI projects more rainfall and less snowfall with minimalistic variations in the annual quantum in future, which underscores the regional variations in the prospective precipitation patterns [38]. This is also different than the historic trends, wherein snowfall significantly reduced by 20% during the 2000s as compared to the 1970s. Quite interestingly, precipitation did insignificantly increase here more during the 1970s than the 1950s, i.e., from 1010 mm/year to 1062 mm/year [19]. This provides evidence of a cyclic phenomena here, where precipitation increased and decreased, and is again likely to increase along the successive climate normals.

**Table 13.** Average precipitations (mm) observed in the western zone (Summerside) during the 2000s and expected in normal years (PoE = 50%) under different projections during the 2030s and 2060s.

| Precipitation (mm) | 1991–2020 | 2021–2050 | | | 2051–2080 | | |
|---|---|---|---|---|---|---|---|
| | Observed | RCP2.6 | RCP4.5 | RCP8.5 | RCP2.6 | RCP4.5 | RCP8.5 |
| January | 69 [b] | 96 [ab] | **106** [a] | **100** [a] | **104** [a] | 97 [ab] | **121** [a] |
| February | 62 [c] | 83 [bc] | **100** [ab] | **91** [ab] | 83 [bc] | **102** [ab] | **114** [a] |
| March | 76 [b] | 99 [ab] | 96 [ab] | **105** [a] | 101 [ab] | 95 [ab] | **107** [a] |
| April | 73 [a] | 87 [a] | 106 [a] | 102 [a] | 90 [a] | 97 [a] | 105 [a] |
| May | 78 [a] | 85 [a] | 91 [a] | 102 [a] | 95 [a] | 84 [a] | 79 [a] |
| June | 85 [a] | 97 [a] | 91 [a] | 80 [a] | 87 [a] | 85 [a] | 86 [a] |
| July | 70 [b] | **104** [a] | 75 [ab] | 93 [ab] | 91 [ab] | **105** [a] | 72 [b] |
| August | 90 [a] | 84 [a] | 86 [a] | 92 [a] | 76 [a] | 88 [a] | 67 [a] |
| September | 80 [a] | 79 [a] | 83 [a] | 80 [a] | 78 [a] | 73 [a] | 62 [a] |
| October | 109 [a] | 96 [a] | 110 [a] | 84 [a] | 94 [a] | 87 [a] | 103 [a] |
| November | 95 [a] | 110 [a] | 102 [a] | 113 [a] | 99 [a] | 120 [a] | 119 [a] |
| December | 91 [a] | 106 [a] | 111 [a] | 113 [a] | 114 [a] | 117 [a] | 114 [a] |
| **Annual** | **978** [b] | **1126** [a] | **1157** [a] | **1154** [a] | **1113** [a] | **1148** [a] | **1149** [a] |

Note: Means that do not share a letter are significantly different at α = 0.05 (95% confidence) as per Tukey's method. Bolded values highlight significantly different ones in the text, the annual are bolded because they represent sum total of all the months.

The interannual uncertainties in precipitation during wet and dry years are provided in Figure 9. Historically, during 1991–2020, the expected precipitations during wet and dry years were 830 and 1145 mm/year, respectively, providing an interannual uncertainty of ~315 mm [19], which is likely to change, and both the dry and wet years would be somewhat wetter, with expected precipitation varying between 1000–1250 mm/year, and 950–1300 mm/year during the 2030s and 2060s, respectively, under RCP4.5. It is also evident from Figure 9 that the temporal distribution of precipitation in dry and wet years would be significantly disturbed. It can therefore be concluded that an increase in precipitation is expected in all dry, normal, and wet years, but uncertainties in its monthly distribution would intensify in the western zone and would pose water management challenges. Likewise, interannual uncertainties were found to increase irrigation requirements

in neighboring southern Quebec two times more in dry years than in normal years, with minimal irrigation requirements in wet years [26].

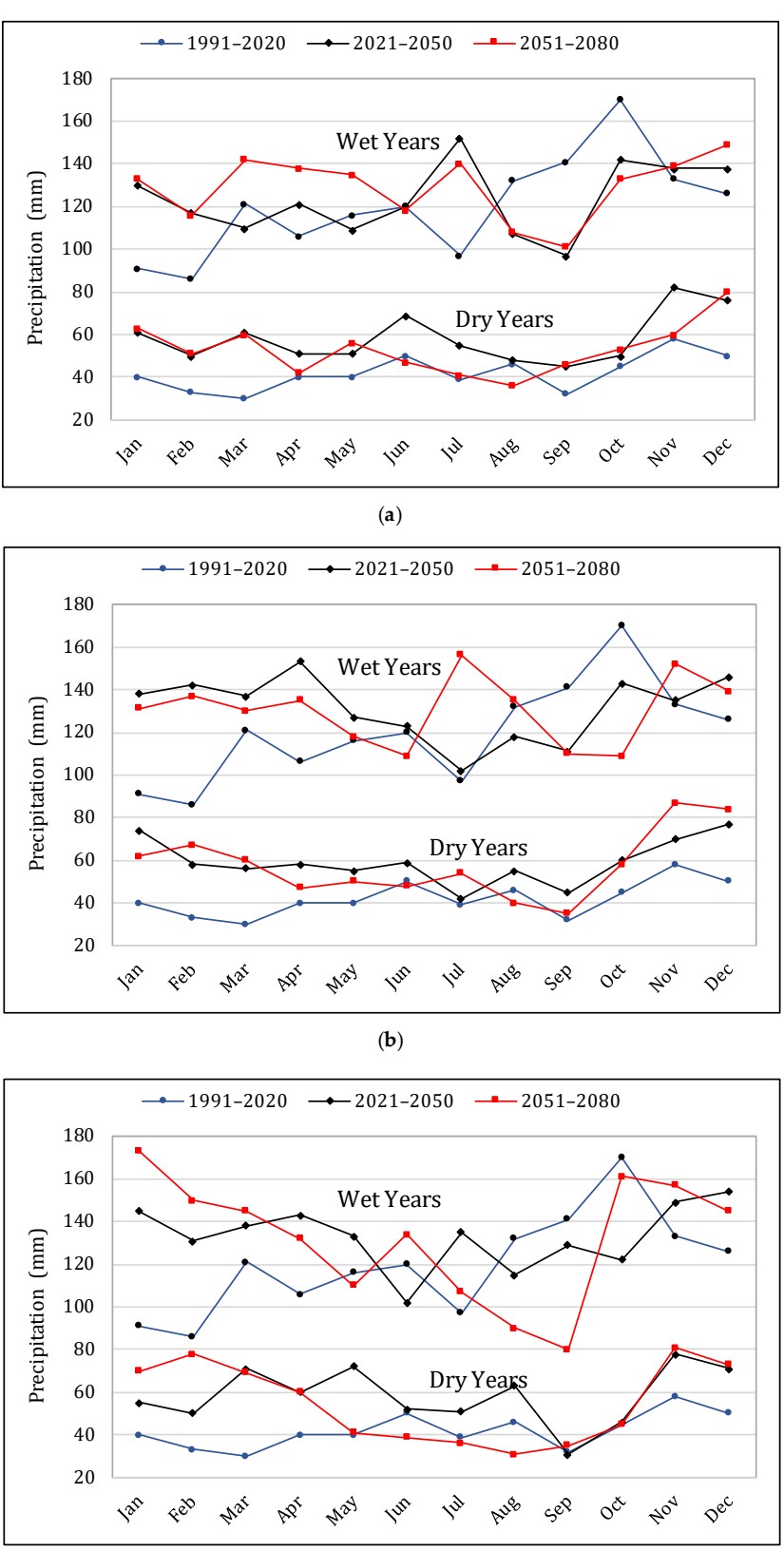

**Figure 9.** Expected monthly precipitations during wet (PoE = 20%) and dry years (PoE = 80%) under the RCP2.6 (**a**), RCP4.5 (**b**), and RCP8.5 (**c**) scenarios in the western zone (Summerside).

The historic observed streamflows of the western zone, i.e., at Carruthers Brook and in Wilmot Valley of the Mill and Wilmot River watersheds, respectively, are given in Table 14 [20]. Just like in the eastern zone, no significant reduction is likely in annual streamflows, though it would reduce from 649 mm/year to 602 mm/year (−7%) during the 2060s under RCP8.5. Climate change impacts on water resources are not intense in wetter regions of the world like PEI; otherwise, in drier regions like the Volta River Basin in West Africa streamflows are likely to reduce by 40% in the next 80 years because of decreased precipitation and increased temperatures [25]. Similarly, in the Soan River Basin of Pakistan, significant reduction in streamflows were recorded during 1983–2012 because of warming and reduced precipitation [11]. As per the model results, direct runoff and baseflow contributions to the streamflows here were 60%, and 40%, respectively. The zone has a relatively higher runoff component than the eastern and central zones because of less-infiltrating soils [17]. More attenuated flows are likely, with more uniform distribution throughout the year. Higher flows in January–March under the effect of elevated temperatures would suppress the annual peak generated in April, and somewhat the one in May. Thereon, flows would again increase during June–August and decrease thereafter. September would be the month of the least flows. In the Wilmot River watershed, the aggregated impacts of climate change and pumping can reduce streamflows 4–13%, and this therefore needs proper planning.

**Table 14.** Average streamflows (mm) observed in the western zone (at Carruthers Brook, and Wilmot Valley of Mill, and Wilmot River watersheds respectively) during the 2000s, and modeled future projections during the 2030s and 2060s.

| Streamflows (mm) | 1991–2020 | 2021–2050 | | | 2051–2080 | | |
|---|---|---|---|---|---|---|---|
| | Observed | RCP2.6 | RCP4.5 | RCP8.5 | RCP2.6 | RCP4.5 | RCP8.5 |
| January | 59 [b] | **87** [a] | **86** [a] | 80 [ab] | 83 [ab] | **86** [a] | **94** [a] |
| February | 42 [b] | **76** [a] | **96** [a] | **77** [a] | **98** [a] | **105** [a] | **97** [a] |
| March | 83 [ab] | 86 [ab] | 92 [ab] | 103 [a] | 69 [b] | 70 [b] | 83 [ab] |
| April | 143 [a] | **49** [b] | **71** [b] | **72** [b] | **56** [b] | **63** [b] | **68** [b] |
| May | 70 [a] | 53 [a] | 58 [a] | 62 [a] | 57 [a] | 56 [a] | 60 [a] |
| June | 34 [b] | 44 [ab] | 47 [ab] | 51 [a] | 45 [ab] | 45 [ab] | 44 [ab] |
| July | 24 [b] | 44 [a] | 35 [ab] | 40 [a] | 34 [ab] | 37 [ab] | 25 [b] |
| August | 19 [ab] | 30 [a] | 19 [ab] | 26 [a] | 22 [a] | 23 [a] | 10 [b] |
| September | 23 [a] | 19 [a] | 14 [ab] | 17 [a] | 14 [ab] | 12 [ab] | 4 [b] |
| October | 35 [a] | 22 [ab] | 29 [ab] | 17 [b] | **18** [b] | **14** [b] | **14** [b] |
| November | 52 [a] | 38 [ab] | 35 [ab] | 32 [b] | 33 [b] | 36 [ab] | 28 [b] |
| December | 65 [a] | 75 [a] | 69 [a] | 69 [a] | 74 [a] | 78 [a] | 75 [a] |
| **Annual** | **649** [a] | **623** [a] | **651** [a] | **646** [a] | **603** [a] | **625** [a] | **602** [a] |
| **Wilmot River** | **652** | (602) | (629) | (623) | (577) | (591) | (571) |

Note: Means that do not share a letter are significantly different at $\alpha = 0.05$ (95% confidence) as per Tukey's method, Note: bracketed values are pumping adjusted as detailed in Section 2.3. Bolded values highlight significantly different ones in the text, the annual are bolded because they represent sum total of all the months.

Model simulated recharges for the Mill and Wilmot River watersheds were weight-averaged to represent the western zone. The historically computed recharges (1995–2014) by the validated model [20] have been compared with the expected prospective recharges under different RCP scenarios in Table 15. It is evident from the table that no significant change is likely in annual recharge in the western zone. Temporally, no significant redistribution is likely during the 2030s, but this would be during the 2060s with more recharge in spring and less in summer. The recharge here is quite resilient against environmental change impacts. Particularly, in the Wilmot River, the recharges either remain minimally changed or can somewhat increase as predicted by Li [22]. It can therefore be concluded that warming would not affect groundwater recharges in this zone because of increasing precipitation and increased recharge induced by warming. This is in accordance with global

projections, wherein wet regions at high latitudes are expected to be hydrologically less affected by climate change [8].

**Table 15.** Average recharge (mm) in the western zone (Mill and Wilmot River watersheds) during 1995–2014 and modeled future projections during the 2030s and 2060s.

| Recharge (mm) | 1995–2014 | 2021–2050 | | | 2051–2080 | | |
|---|---|---|---|---|---|---|---|
| | Historical | RCP2.6 | RCP4.5 | RCP8.5 | RCP2.6 | RCP4.5 | RCP8.5 |
| January | 23 ab | 22 b | 28 ab | 24 ab | 25 ab | 26 ab | 34 a |
| February | 16 bcd | 12 d | 15 bcd | 12 cd | 21 bc | 22 b | **35 a** |
| March | 18 c | 25 c | 24 c | 29 c | **42 b** | **53 a** | 58 a |
| April | 31 b | 44 ab | 45 ab | **52 a** | 43 ab | 46 ab | 51 a |
| May | 45 a | 41 a | 42 a | 44 a | 38 a | 38 a | 43 a |
| June | 38 ab | 29 abc | 30 abc | 35 a | 28 abc | 26 bc | 26 c |
| July | 32 a | 25 ab | 21 abc | 22 abc | **19 bc** | **18 bc** | **16 c** |
| August | 25 a | 20 ab | **12 cd** | **14 bc** | **14 bcd** | **13 bcd** | **7 d** |
| September | 20 a | 12 ab | **8 bc** | **10 b** | **9 bc** | **8 bc** | **3 c** |
| October | 20 a | 13 a | 13 a | 10 ab | 12 ab | 9 ab | **4 b** |
| November | 24 a | 26 a | 28 a | 22 a | 23 a | 22 a | 17 a |
| December | 29 a | 37 a | 41 a | 37 a | 33 a | 37 a | 35 a |
| **Annual** | **323 a** | **307 a** | **313 a** | **315 a** | **307 a** | **312 a** | **326 a** |
| **Wilmot River** | 345 | 335 | 343 | 341 | 336 | 343 | 362 |

Note: Means that do not share a letter are significantly different at $\alpha$ = 0.05 (95% confidence) as per Tukey's method. Bolded values highlight significantly different ones in the text, the annual are bolded because they represent sum total of all the months.

## 4. Island Perspective and Conclusions

A summarized comparison of observed hydrological parameters during 1991–2020 (1995–2014 for recharge), and future projections for 2021–2050 (2030s), and 2051–2080 (2060s) under the moderate emission scenario (RCP4.5) are given in Table 16. Western PEI was found to be the most resilient against environmental change impacts, wherein a significant increase in precipitation, from 978 mm/year to 1150 mm/year (+17%), is likely to offset the significant warming effects. Warming would cause the present (1995–2014) ET of 460 mm/year to increase by 12% during 2021–2080. Overall, minimal effect on water availability is likely, and streamflows (~650 mm/year) or recharges (~320 mm/year) would almost remain the same, but their temporal redistribution would affect seasonal water availability. Nevertheless, a rise in pumping can reduce streamflows in the Wilmot River watershed up to 9% during the 2030s and 13% during the 2060s. Precipitation expectancy during dry and wet years would change from the historic 830–1145 mm/year to 1000–1250 mm/year, and to 950–1300 mm/year during the 2030s and 2060s, respectively, under RCP4.5. This indicates that both the dry and wet years would become somewhat wetter than before. Despite increasing precipitation, and stable streamflows and recharges, inter-annual uncertainty in precipitation during dry and wet years, its temporal redistribution and increases in rainfall intensity may pose some water management challenges.

In the eastern zone, no change in precipitation is likely, in contrast with the historic trends wherein it has been continuously increasing since the 1940s. Significant warming is expected; moreover, the eastern zone would be warmer than the central and western zones. Even then, it seems resilient and minimal impacts on current streamflows (692 mm/year) and recharges (597 mm/year) are likely, but with temporal redistribution, as evident from Table 16. Relatively higher streamflows and recharges in the zone are due to the fact that it is almost entirely under forest. Prospectively, early snowmelt during spring and thus more attenuated contribution to streamflow and recharges are the underlying reasons of stability. Pumping (1.3 mm/year) is not likely to impact the water balance, as per the existing and projected trends. During 1995–2014, ET was 474 mm/year, which would increase 4% during 2021–2080. Therefore, water availability would be minimally affected by up to −8%

because of increased ET, etc. Moreover, uncertainty in precipitation expectancy between dry and wet years is ~300 mm/year, and both the dry and wet years would become somewhat wetter. All that, and the monthly distribution of hydrological parameters, may be looked at for sustainable water management.

**Table 16.** Observed and projected (RCP4.5) climatic and hydrological parameters in different zones of Prince Edward Island.

| Parameter | Eastern PEI | | | Central PEI | | | Western PEI | | |
|---|---|---|---|---|---|---|---|---|---|
| | 2000s | 2030s | 2060s | 2000s | 2030s | 2060s | 2000s | 2030s | 2060s |
| Temperature (°C) | 6.86 [a] | **8.62 [b]** | 9.71 [c] | 5.90 [a] | **8.46 [b]** | 9.59 [c] | 6.22 [a] | **8.37 [b]** | 9.54 [c] |
| Precipitation (mm/year) | 1182 [a] | 1192 [a] | 1180 [a] | 1141 [a] | 1198 [a] | 1192 [a] | 985 [b] | **1157 [a]** | **1148 [a]** |
| Streamflows (mm/year) | 692 [a] | 688 [a] | 663 [a] | 737 [a] | 680 [a] | 649 [a] | 649 [a] | 651 [a] | 625 [a] |
| | | | | 565 [1] | 481 [1] | 448 [1] | 652 [2] | 629 [2] | 591 [2] |
| Recharge (mm/year) | 597 [a] | 536 [a] | 539 [a] | 446 [a] | 419 [a] | 441 [a] | 323 [a] | 313 [a] | 312 [a] |
| | | | | 504 [1] | 413 [1] | 413 [1] | 345 [2] | 343 [2] | 343 [2] |

Note: Means that do not share a letter are significantly different at $\alpha = 0.05$ (95% confidence) as per Tukey's method (bolded), [1] values for the Winter River watershed (streamflows are pumping adjusted), and [2] values for the Wilmot River watershed (streamflows are pumping adjusted).

Warming in the central zone would be statistically significant, but precipitation is likely to increase insignificantly by 5% (~50 mm/year). Nevertheless, streamflows and recharges in this zone were found to be vulnerable because of the environmental change impacts of varied climate and pumping. The zone has steep topography and the highest hydraulic conductivity, population concentration, and pumping, which are underlying reasons of its vulnerability. It is evident from the table that streamflows (737 mm/year) and recharges (446 mm/year) are likely to reduce insignificantly; nevertheless, streamflows could reduce under RCP8.5 during the 2060s by 16%. In the high-water use Winter River watershed, even with no further increase in pumping, streamflows could reduce up to 24% under RCP8.5 during the 2060s, and recharge by 16% because of warming, higher ET thereof, and other changes.

The island is surrounded by the Gulf of St. Lawrence, which moderates the intensity of both hotness and coldness. Therefore, warming may have some beneficial impact on the island climate, e.g., relatively longer growing seasons, moderated colds, narrowed inter-annual uncertainty in precipitation during dry and wet years, and more evenly distributed streamflows and recharges, etc. However, that temporal redistribution of precipitation, increase in rainfall intensity, population concentration, and pumping are some water management challenges, which need careful planning and management.

**Author Contributions:** Conceptualization, A.Z.B., A.A.F., N.K., W.P. and B.A.; investigation, A.Z.B., A.A.F. and Q.L.; validation, A.A.F., N.K., W.P. and B.A.; formal analysis, A.Z.B., A.A.F. and Q.L.; resources, A.A.F., N.K., W.P. and B.A.; data curation, A.Z.B., A.A.F., W.P. and Q.L.; writing—original draft preparation, A.Z.B. and A.A.F.; writing—review and editing; Q.L., N.K., W.P. and B.A.; supervision, A.A.F. and N.K.; project administration, A.A.F. and N.K.; funding acquisition, A.A.F. All authors have read and agreed to the published version of the manuscript.

**Funding:** The Natural Science and Engineering Research Council of Canada (NSERC), and the PEI Department of Environment, Energy, and Climate Action provided funding for this study.

**Data Availability Statement:** The data presented in this study are available on request from the corresponding authors. The data are not yet publicly available.

**Acknowledgments:** The support extended by the Faculty of Sustainable Design Engineering at the University of Prince Edward Island is acknowledged.

**Conflicts of Interest:** The authors declare no conflict of interest.

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
