# Peer review of "Prospective Climates, and Water Availabilities under Different Projections of Environmental Changes in Prince Edward Island, Canada"

_water, doi:10.3390/w14050740_

Round 1

Reviewer 1 Report

1. The Abstract is too lengthy In first few lines authors presented details which are already well known. It is suggested to revise abstract and present main aim and significant findings of this study in General.  2. In introduction section when authors discussed impact of climate change and human activities on water resources only limited studies were cited in literature. However, these impacts are common at Global scale therefore it is suggested to include studies from different parts of the world in literature so that this study could be more useful for global readers. Following contributions can be considered in this regard:  i. Attribution of runoff change in the alpine basin: a case study of the Heihe Upstream Basin, China ii. Understanding the impacts of climate change and human activities on streamflow: a case study of the Soan River basin, Pakistan 2. In material and method section authors added too much details from previous studies and it is not clear that what are main contributions from authors? It is suggested to add flow chart of methodology and justify main contributions. 3. It is suggested to re-organise your results and significant findings authors mixed results and discussions.  4. Literature cited in introduction section should be used for discussion.  

Reviewer 2 Report

The topic of the article is interesting and up-to-date. The paper fits within the aims and scope of the Journal. In my opinion, the work contains the following flaws and shortcomings:

  1. The authors limited their perspective to a case study with no literature background in the introduction and a discussion referenced to other regions/ case studies.  The introduction is very narrow, with a significant part dealing with the description of the study area. A broad presentation of the research problem and methodology against the results/works of other authors, from other regions, is necessary. Without this, the results are not reliable. 
  2. The division into time intervals - 2030s and 2060s  is not logical and clear. In the text, the Authors use both designations anyway - 1991-2020 23 (2000s), 2021-2050 (2030s), 2051-2080s (2060s). It should be clarified.
  3. The text contains numerous editorial errors, for example: line 19 - under - lower case at the beginning of the sentence. Line 147 0.10-1.20  C. Line 188 please remove i.e. Line 203 and 287 - unnecessary dot. Line 251 and 613 reference cited by name of author and year of publication...

Round 2

Reviewer 2 Report

The authors have considered most of the reviewer's comments. The article is suitable for publication.